# Revised records of atmospheric trace gases $CO_2$, $CH_4$, $N_2O$ and $\delta^{13}C$-$CO_2$ over the last 2000 years from Law Dome, Antarctica

Mauro Rubino[1,2], David M. Etheridge[2], David P. Thornton[2], Russell Howden[2], Colin E. Allison[2], Roger J. Francey[2], Ray L. Langenfelds[2], L. Paul Steele[2], Cathy M. Trudinger[2], Darren A. Spencer[2], Mark A. J. Curran[3,4], Tas D. Van Ommen[3,4], Andrew M. Smith[5]

[1]School of Geography, Geology and the Environment, University of Keele, ST5 5BG, UK
[2]Climate Science Centre, CSIRO Oceans and Atmosphere, Aspendale, Victoria, 3195 Australia
[3]Antarctic Climate and Ecosystems Cooperative Research Centre, University of Tasmania, Private Bag 80, Hobart, Tasmania 7005, Australia
[4]Australian Antarctic Division, Channel Highway, Kingston, Tasmania 7050, Australia
[5]Australian Nuclear Science and Technology Organisation (ANSTO), Locked Bag 2001, Kirrawee, NSW 2232, Australia

*Correspondence to*: Mauro Rubino (m.rubino@keele.ac.uk)

**Abstract.** Ice core records of the major atmospheric greenhouse gases ($CO_2$, $CH_4$, $N_2O$) and their isotopologues covering recent centuries provide evidence of biogeochemical variations during the Late-Holocene and Pre-Industrial Periods and over the transition to the Industrial Period. These records come from a number of ice core and firn air sites, and have been measured in several laboratories around the world and show common features, but also unresolved differences. Here we present revised records, including new measurements, performed at the CSIRO Ice Core Extraction LABoratory (ICELAB) on air samples from ice obtained at the high accumulation site of Law Dome (East Antarctica). We are motivated by the increasing use of the records by the scientific community and by recent data-handling developments at CSIRO-ICELAB. A number of cores and firn air samples have been collected at Law Dome to provide high-resolution records overlapping recent, direct atmospheric observations. The records have been updated through a dynamic link to the calibration scales used in the Global Atmospheric Sampling LABoratory (GASLAB) at CSIRO, which are periodically revised with information from the latest calibration experiments. The gas-age scales have been revised based on new ice-age scales, and the information derived from a new version of the CSIRO firn diffusion model. Additionally, the records have been revised with new, rule-based selection criteria and updated corrections for biases associated with the extraction procedure, and the effects of gravity and diffusion in the firn. All measurements carried out in ICELAB-GASLAB over the last 25 years are now managed through a database (the ICElab dataBASE or ICEBASE) which provides consistent data management, automatic corrections and selection of measurements, and a web-based user interface for data extraction. We present the new records, discuss their strengths and limitations and summarise their main features. The records reveal changes in the carbon cycle and atmospheric chemistry over the last two millennia, including the major changes of the anthropogenic era and the smaller, mainly natural variations beforehand. They provide the historical data to calibrate and test the next inter-comparison of models used to predict future climate change (Coupled Model Inter-comparison Project - phase 6, CMIP6). The datasets described in this paper, including spline fits, are available at https://doi.org/10.25919/5bfe29ff807fb (Rubino et al., 2018).

# 1 Introduction

The three well-mixed (long-lived) atmospheric greenhouse gases (GHGs) that contribute the most to current global warming are $CO_2$, $CH_4$ and $N_2O$. Their concentrations have been increasing since the beginning of the Industrial Period, causing most of the current ~1 °C temperature increase above the average global temperature in the period 1861-1880 (Stocker et al., 2013). The temperature increase limit of 2 °C set by the Paris Agreement for 2100 requires substantial reduction of GHG emissions in the next decades, and, consequently, significant reductions in the rates of GHG concentration increases. Predicting how GHG concentrations will vary in the future requires a clear understanding of the biogeochemical processes responsible for their variations. However, models of future long-term climate changes predict a large range in GHG concentrations for a given scenario of emissions (Friedlingstein et al., 2014), and one of the key uncertainties is associated with feedbacks in the coupled carbon-climate system (Arora et al., 2013). Climate modellers have analysed and compared results from state-of-the-art climate model simulations to gain insights into the processes of climate variability, change and feedbacks through the Coupled Model Inter-comparison Project (CMIP). In CMIP, records of GHGs can be used either as forcing or as a diagnostic (Graven et al., 2017; Meinshausen et al., 2017). However, real-time records of GHGs started in a period when anthropogenic forcing was already very significant, and the atmosphere and the Earth system were in strong disequilibrium, and therefore do not provide a balanced state for model spin-up. Additionally, temperature and $CO_2$ have both increased almost continuously through the 20th century, making it difficult to separate the impacts of $CO_2$ on carbon sinks from the impacts of temperature increase on these sinks. Furthermore, real-time records are often too short to draw strong conclusions on multi-decadal variability. To provide a balanced system for model spin-up, and evaluate the ability of models to capture observed variability on multi-decadal and longer time-scales, a branch of CMIP ('Historical Simulations') starts in 1850 (Eyring et al., 2016), while another branch (the 'Paleoclimate Modelling Intercomparison Project', PMIP) focuses on paleo-climate simulations (Schmidt et al., 2014). Yet, policymakers need short-term predictions of global warming (next decades to century), and the Intergovernmental Panel on Climate Change has very recently provided a special report on the impacts of global warming of 1.5 °C above Pre-Industrial levels. The last millennium is a very suitable period to support these types of investigations since the Earth system was much closer to its current state than e.g. previous periods of glacial-interglacial transition.

Ice cores are exceptional archives of factors influencing past climate change because they contain a large range of substances, including water ($H_2O$) in the ice itself, as well as ionic species, organic molecules and atmospheric gases sealed in bubbles (Barbante et al., 2010). They can span from polar (Antarctica and Greenland mostly) to tropical (high altitude) sites (Thompson et al., 2013), and extend several hundred thousand years back in time (Higgins et al., 2015; Wolff et al., 2010). Ice cores from different locations have different accumulation rates and temperatures, which translate into differences in time resolution, the age of the deepest layers and archive suitability. Focusing on the last millennium, multiple ice core records of GHG concentration and isotopic composition are available:

- $CO_2$ from EDML (EPICA Dronning Maud Land, Antarctica) and South Pole (Siegenthaler et al., 2005), Law Dome, East Antarctica (Etheridge et al., 1996; MacFarling Meure et al., 2006; Rubino et al., 2013), DML (Dronning Maud Land, Rubino et al., 2016) and WAIS (West Antarctic Ice Sheet, Ahn et al., 2012);

- $\delta^{13}C$-$CO_2$ from Law Dome (Francey et al., 1999; Rubino et al., 2013), WAIS (Bauska et al., 2015) and DML (Rubino et al., 2016);

- $CH_4$ from NEEM (Rhodes et al., 2013) and from GISP2 (Mitchell et al., 2013) in Greenland, Law Dome (Etheridge et al., 1998; MacFarling Meure et al., 2006) and WAIS (Mitchell et al., 2011) in Antarctica;

- $N_2O$ from EUROCORE and GRIP in Greenland (Flückiger et al., 1999), Dome C (Flückiger et al., 2002), and Law Dome (MacFarling Meure et al., 2006) in Antarctica.

There are other records focusing on periods other than the last centuries, but also covering the whole or part of the Industrial and the Pre-Industrial Periods (i.e. for $N_2O$: H15 by Machida et al., 1995; Styx glacier by Ryu et al., 2018; Talos Dome by Schilt et al., 2010). We have decided not to include them in our comparison because their temporal resolution (Schilt et al., 2010) and/or coverage (Machida et al., 1995; Ryu et al., 2018) limits their value for comparison with the records focusing on the last centuries.

There are real differences between records of the same GHG from different sites caused by atmospheric features, such as the inter-hemispheric gradient (North-South or Greenland vs Antarctica). The inter-hemispheric gradient is different from one GHG to another, depending on the balance between, and the distribution of, sources and sinks for that specific GHG in the two hemispheres, as well as on the atmospheric circulation and the atmospheric lifetimes of the gases. There are also differences which do not reflect atmospheric changes, due, for example, to the characteristics of the sites where the ice is sampled. Ice core site characteristics influence the measured gas records due to the gaseous diffusion through the upper-most layers of porous, compacting snow, the 'firn' (Schwander et al., 1993). Together, diffusion in firn and gradual bubble close-off result in a smoothed representation of the atmospheric history in ice core gas records. The smoothing process depends on the depth of the firn layer and on how quickly bubbles close off and trap air during firn to ice transition. Ice cores sites in Greenland generally have higher accumulation rates and temperatures than in Antarctica. Consequently, GHG records from many Antarctic sites are usually a more smoothed representation of the atmospheric history. Unfortunately, there is no reliable $CO_2$ record available from Greenland because there is evidence of *in-situ* production of $CO_2$ (Anklin et al., 1995; Barnola et al., 1995). The most likely explanation for this is a high level of impurities in Greenland ice reacting with acidity and/or hydrogen peroxide (Jenk et al., 2012; Tschumi and Stauffer, 2000). Law Dome, Antarctica, provides the best time-resolved ice core records due to the very high accumulation rate at this site (Etheridge et al., 1996; Goodwin, 1990), even more so than Greenland. Also, records from multiple Law Dome sites show no evidence of *in-situ* production because they agree with records from colder sites in Antarctica (Rubino et al., 2016; Siegenthaler et al., 2005), and compare closely with each other, with air extracted from the firn, and with modern atmospheric records (Rubino et al., 2013).

However, there are unexplained differences between records of the same GHG, particularly for $CO_2$. For example, while the $CO_2$ records of South Pole and EDML over the last centuries are consistent with Law Dome when their broader age

smoothing is taken into account (Rubino et al., 2016; Siegenthaler et al., 2005), the WAIS $CO_2$ record is on average 3 ppm higher than the Law Dome $CO_2$ record (Ahn et al., 2012). The similarity between the high frequency variations of the $CH_4$ records from Law Dome and WAIS (Mitchell et al., 2011) suggests that the two sites (Law Dome and WAIS) introduce similar smoothing of the atmospheric signals. However, the Law Dome $CO_2$ minimum measured around 1610 CE does not have a corresponding feature at WAIS (Ahn et al., 2012). Considering that a comparison between the two laboratories where Law Dome and WAIS samples were measured has shown no significant offset (Ahn et al., 2012), the differences between the WAIS and the Law Dome $CO_2$ records could be explained by a small effect of *in-situ* production at WAIS. Additionally, there is a significant difference in the mean Pre-Industrial level of $\delta^{13}C$-$CO_2$ measured at WAIS and Law Dome (Bauska et al., 2015; Rubino et al., 2016). These differences need to be resolved with inter-calibration campaigns between different laboratories, using ice cores from different sites (including new high accumulation cores) and accurate modelling of gas-age (both the mean value and spread) at each site.

To provide the most consistent datasets possible for the past centuries, we have previously compared the Law Dome records of $CO_2$, $CH_4$, $N_2O$ and $\delta^{13}C$-$CO_2$ to firn and modern atmospheric measurements (MacFarling Meure et al., 2006; Rubino et al., 2013). The consistency between these measurements is evidence of our ability to extend current atmospheric records back in time using ice and firn. However, because of the emissions during the Industrial Revolution, our measurements of modern and old (Pre-Industrial) air samples lie in different concentration ranges and the calibrations used for measurements of modern air samples are, therefore, in a concentration range rather different from that used for measurements of old air samples. The measurements performed in ICELAB-GASLAB at CSIRO have the advantage of being calibrated across the range of concentrations of old and modern air sample measurements. Also, ice core gas extractions and analyses are technically challenging, and different people at CSIRO-ICELAB have produced those measurements over almost three decades. Thus, it is possible that the extraction/analysis procedures have introduced different biases over time, influencing the measurements by variable amounts. However, except for minor developments over time (Etheridge et al., 1996; Francey et al., 1999; MacFarling Meure et al., 2006; Rubino et al., 2013, 2016) the equipment used for extraction and analysis has not fundamentally changed.

In this study, we describe the procedure recently developed at CSIRO ICELAB-GASLAB to perform calibration scale updates and data selection and correction automatically and in a consistent way for all measurements made over the last 25 years. In the Supplement, we provide a detailed explanation of the database recently created to store, process and extract the information about the samples analysed, the measurements performed and the results obtained. We present updated records of $CO_2$, $CH_4$, $N_2O$ and $\delta^{13}C$-$CO_2$ measured in ice and firn air from Law Dome (Rubino et al., 2018). After merging them with other relevant records, they will be used to run models participating at CMIP6 (Graven et al., 2017; Meinshausen et al., 2017). We discuss the strengths and limitations of the Law Dome GHG records and compare our records with other records from different sites to show similarities and unresolved discrepancies. Finally, we discuss the main features of those records, their implications for biogeochemical, atmospheric and climatic studies, and possible future lines of research.

## 2 Methods

### 2.1 Law Dome

The ice cores used in this study, referred to as DE08, DE08-2, DSS and DSS0506, were drilled at Law Dome, East Antarctica (Fig. 1). Law Dome is a relatively small (~150 km diameter and 1390 m high) ice sheet on the coast of Wilkes Land. It receives large and regular snowfall mainly from the east, and the surface rarely melts in the colder central regions. The ice flow is mainly independent of the flow of the main East Antarctic ice sheet because of the drainage around Law Dome by two glacier systems (the Totten and the Vanderford). Re-working of the accumulated snow is insufficient to erase annual layers as high wind speeds are relatively infrequent. The resulting annual layering is thick and regular and preserved for much of the ice thickness (van Ommen et al., 2005).

DE08 and DE08-2 were drilled in 1987 and 1993, respectively, only 300 m apart and 16 km east of the summit of Law Dome (66°44'S, 112°50'E, 1390 m above mean annual sea level), and have an accumulation rate of approximately 1100 kg m$^{-2}$ yr$^{-1}$ (equivalent to 1.4 m of ice per year). DSS (Dome Summit South) was drilled between 1988 and 1993, 4.6 km south-south west of the summit, and has an accumulation rate of about 600 kg m$^{-2}$ yr$^{-1}$ (Etheridge et al., 1996; Goodwin, 1990; van Ommen et al., 2005). In January-February 1993, air was sampled from the firn layer at DE08-2, providing air with mean ages back to 1976 A.D. (Etheridge et al., 1996). Another firn campaign was carried out at DSSW20K (accumulation rate of approximately 150 kg m$^{-2}$ yr$^{-1}$), 20 km west of DSS in December 1997 (Sturrock et al., 2002), which provided air dating back to about 1940 A.D. (Trudinger et al., 2002b). DSS0506 was thermally drilled in a dry hole (Burn-Nunes et al., 2011) during the 2005/2006 austral summer near the Law Dome summit (66°46'S, 112°48'E, 1370 masl). The site has an accumulation rate of about 600 kg m$^{-2}$ yr$^{-1}$ and a mean annual temperature of -22 °C.

### 2.2 ICELAB extraction

Measurement of the composition of air in ice core bubbles requires an extraction step to release the air from ice. The dry extraction technique used at ICELAB has been described in detail in previous publications (Etheridge et al., 1996; MacFarling Meure et al., 2006), with recent minor alterations to optimise extraction and measurement of $\delta^{13}$C-CO$_2$ analyses (Rubino et al., 2013). Briefly, after ice sample selection and preparation (removing the outer 5–20 mm with a band saw), typically 0.7–1.3 kg of ice is placed in a polyethylene bag (Layflat, USP®) and cooled down to -80 °C in a chest freezer for at least 24 h prior to extraction. The ice is then placed inside a perforated inner cylinder ('cheese grater') fixed inside an internally electropolished stainless steel container which is then evacuated to less than 10$^{-4}$ Torr and maintained at that pressure for at least 25 min. The ice is then grated by mechanically shaking the container for 10 minutes, which releases the trapped air. The process yields on average 70 mL (range 50-90 mL) of air, estimated from the pressure in the extraction line (whose volume has been previously calibrated). The air is passed through a water vapour trap (~-100 °C) and then cryogenically collected in an electropolished and preconditioned stainless steel trap at around 20 K (-253 °C). The sample trap is warmed in a water bath at room temperature (~25 °C) for 5 min to vaporise and mix the gases before being

transported into the instrument laboratory. Samples are analysed on gas chromatographs (GCs) for $CO_2$, $CH_4$, $CO$, $H_2$, and $N_2O$ within 24 h after extraction, and on the isotope ratio mass spectrometer (IRMS) for $\delta^{13}C$ and $\delta^{18}O$ within 12 h.

To estimate the uncertainty and any possible bias introduced by the extraction procedure (called the blank correction), test samples are run together with the real ice samples. The test samples can either be reference air samples of known composition processed with no ice present (named 'blanks'), or reference air samples injected over ice grown with no visible bubbles in it and grated as for an actual ice core sample (the so called bubble free ice, or BFI). BFI is grown in ICELAB by keeping a container filled with deionized water in thermal equilibrium, in order to grow ice as slowly as possible from the bottom to the top of the container. The container features Plexiglass sidewalls that are electrically heated. The water exchanges heat only through the metallic base and freezes from the bottom to the top. If the process is slow enough, the produced ice is free of visible bubbles. The results of the tests performed on ICELAB-BFI, as well as on other, externally grown BFI, have been extensively described by Rubino et al. (2013).

## 2.3 GASLAB analysis

Each extracted air sample is analysed for trace gas concentrations (defined as mole fractions in parts per million (ppm) or parts per billion (ppb) in dry air) using several GCs in GASLAB. A Series 400 CARLE/EG&G (Tulsa, Oklahoma, USA) GC equipped with flame ionization detector is used to measure $CH_4$ and $CO_2$ (the latter converted, after column separation, to $CH_4$ using a nickel catalyst at 400 °C). A Trace AnalyticalRGA3 (Menlo Park, California, USA) GC, equipped with a mercuric oxide reduction gas detector, is used to measure $CO$ and $H_2$, which reduces HgO to gaseous Hg for detection by UV absorption. $N_2O$ is measured on a Shimadzu GC-8AIE (Kyoto, Japan) equipped with an electron capture detector. In normal GASLAB operation, air samples (including those sampled from firn) in low-pressure flasks and high-pressure cylinders are injected and analysed on the GCs using automated inlet systems to ensure reproducibility and minimum sample consumption (Francey et al., 2003). Because of the limited amount of air available, a semi-automated procedure is used to inject the small volume of ice core air into the GCs inlet systems. Approximately 15-20 mL are used to measure $CO_2$, $CH_4$, $CO$ and $H_2$, and 12-15 mL are used to measure $N_2O$. The remaining air (typically 40 mL) is used for $\delta^{13}C$ and $\delta^{18}O$ measurements. The volumes indicated are total volume used for flushing gas transfer lines as well as for analysis.

The $\delta^{13}C$ and $\delta^{18}O$ of the $CO_2$ in the residual air are measured using the MAT252 (Finnigan MAT GmbH, Bremen) IRMS located in GASLAB. Low pressure, large volume whole air samples from flasks (atmospheric or firn air samples) and the small volume, high pressure, whole air samples from ICELAB are introduced into the IRMS through a common inlet (multiport) equipped with an all stainless steel mass flow controller (Brooks 5850) to ensure constant mass flow conditions for all samples. The IRMS uses two cryogenic traps (MT Box C, Finnigan) to retain water vapour and other condensable gases and to extract $CO_2$ (plus $N_2O$) from air. The residual $CO_2$ (and $N_2O$) is injected into the MAT252 ion source via a dedicated micro-volume and crimped capillary. Nitrous oxide ($N_2O$) has identical molecular masses to $CO_2$ and interferes with the isotopic analyses. To remove this interference, a correction is made to the IRMS output in GASLAB using the relative ionisation efficiency of $N_2O$ and $CO_2$, the isotopic composition of $N_2O$ and the measured $N_2O$ and $CO_2$

concentration, as described in detail by Allison and Francey (2007). High precision isotopic ratios are determined by alternating sample $CO_2$ and reference $CO_2$ injected via matched crimped capillaries. The carbon isotopic ratio of the sample (sa) is expressed relative to the reference (ref) following Eq. (1):

$$\delta^{13}C = \left[ \frac{(^{13}C/^{12}C)_{sa}}{(^{13}C/^{12}C)_{ref}} - 1 \right] \times 1000 \tag{1}$$

When comparing measurements performed more than 20 years apart, rigorous traceability in the propagation of calibration scales becomes an important factor. This is obtained with a long-term, continuous comparison of standard cylinders for both GC (Francey et al., 2003) and IRMS (Allison and Francey, 2007) analyses. The current calibration scales used are WMO X2004A for $CH_4$, WMO X2007 for $CO_2$, NOAA 2006A for $N_2O$ and CSIRO2005 for $CO_2$ isotopes ($CO_2$-in-air scale, which is linked to the VPDB-$CO_2$ scale).

**2.4 ICELAB database**

A new database allows storage, selection, correction, updating and extraction of the data produced in ICELAB-GASLAB. It allows results to be dynamically updated if changes in analytical methods or calibration scales are implemented, keeping ice, firn and atmospheric measurements consistent with each other. Data are stored in tables where the information associated with each specific sample is linked via a Universal Analysis Number (UAN) that acts as the index for combining all

information. The structure of the database and its tables are described in detail in the Supplement. The database includes procedures, which automatically perform sample selection, correct results and estimate uncertainty, and provide a routine for data extraction (see Supplement for details).

**3 Results and discussions**

**3.1 Development of the Law Dome GHG records, internal consistency, unexplained discrepancies and limitations**

The first Pre-Industrial record of $CO_2$ from Law Dome covering the whole last millennium was published by Etheridge et al. (1996, reported here with red squares in Fig. 2a and 3a). It was one of the first ice core records to show the overlap with firn (Fig. 3a) and contemporary atmospheric measurements. The overlap of ice core and contemporary atmospheric measurements is one of the main advantages of the Law Dome ice core sites, due to their high snow accumulation rates and the resultant relatively quick bubble close-off time and recently enclosed air. This feature has been described extensively in

previous papers (Etheridge et al., 1996; MacFarling Meure et al., 2006; Rubino et al., 2013) and used, together with the overlap between different cores, to demonstrate our confidence in extending contemporary GHG concentration measurements back in time. Based on replicate analyses of test samples (blanks and BFI) over time periods of several months, on ice samples within an annual layer, and the overlap mentioned above, Etheridge et al. (1996) estimated that the uncertainty of the $CO_2$ measurements was 1.2 ppm ($1\sigma$). The major biogeochemical events discussed in Etheridge et al.

(1996) were the LIA (Little Ice Age) $CO_2$ decline between 1550 and 1750 CE with the subsequent recovery from the LIA perturbation between 1750 and 1800 CE in the Pre-Industrial Period, and the 1940s stabilisation of atmospheric $CO_2$ concentration (Fig. 3a) which ended just before the Mauna Loa and South Pole atmospheric records began.

A few years later, the same authors published the Law Dome Pre-Industrial record of $CH_4$ covering the last millennium (Etheridge et al., 1998, red squares in Fig. 2b and 3b). The tight overlap for the first time between ice, firn and contemporary atmospheric $CH_4$ measurements that began more than 20 years later than for $CO_2$, confirmed that the ice core air record is a faithful representation of the past atmospheric $CH_4$ concentration (Fig. 3b). The estimated uncertainty was 5 ppb. The major features discussed in Etheridge et al. (1998) were the LIA $CH_4$ decline, supporting a terrestrial origin for the synchronous $CH_4/CO_2$ decrease, and the rapid increase in $CH_4$ growth rates after 1945 CE which peaked in 1981 CE, just as atmospheric monitoring began. It was also possible to determine the Pre-Industrial inter-hemispheric difference in $CH_4$ (24-58 ± 10 ppb), based on comparison with $CH_4$ measurements from Greenland (Eurocore and GISP2), also made in ICELAB/GASLAB, and supporting evidence from Blunier et al. (1993) and Chappellaz et al. (1997). The variability over time of the $CH_4$ Pre-Industrial inter-hemispheric gradient provides a constraint to quantify variations of sources and sinks of $CH_4$ (Mitchell et al., 2013). The same is not possible for $CO_2$ because of the above-mentioned *in-situ* production in Greenland ice.

To quantify variations of the sources and sinks of $CO_2$, Francey et al. (1999) measured its isotopic ratio ($\delta^{13}C$, red squares in Fig. 2c and 3c) in Law Dome ice. This record provided a means to quantify the relative $CO_2$ uptake by the land and the ocean to the total atmospheric $CO_2$ change (Trudinger et al., 2002a), when the emissions from fossil fuel and land use change were taken into account for the Industrial Period, and assuming that, in the Pre-Industrial Period, there was no significant influence of anthropogenic activities on the atmospheric $\delta^{13}C$-$CO_2$ (Pongratz and Caldeira, 2012; Stocker et al., 2011), and also assuming no significant changes in inter-hemispheric $CO_2$ exchange times (Francey and Frederiksen, 2016; Frederiksen and Francey, 2018). The $\delta^{13}C$-$CO_2$ decrease measured by Francey et al. (1999) in the last two centuries (Fig. 3c) is mainly due to $^{13}C$-depleted $CO_2$ derived from fossil fuel $CO_2$ emissions, and is important evidence of the prominent role of anthropogenic emissions on the Industrial Period $CO_2$ increase. Francey et al.(1999) also discussed the increase of $\delta^{13}C$-$CO_2$ during the LIA, supporting the interpretation of a terrestrial origin for the synchronous $CH_4/CO_2$ decrease (Trudinger et al., 1999), though with lower sampling resolution compared to the $CO_2$ in Etheridge et al. (1996). Francey et al. (1999) estimated statistical and systematic $\delta^{13}C$-$CO_2$ biases between 0.025 and 0.07 ‰, and uncertainties of up to ±0.05 ‰, but found an unexplained discrepancy of up to 0.2 ‰ (Trudinger, 2000, section 3.8) around 1900 CE from the South Pole $\delta^{13}C$-$CO_2$ firn record measured at NOAA-INSTAAR (National Oceanic and Atmospheric Administration-Institute of Arctic and Alpine Research, Boulder, Colorado).

The early Law Dome GHG records have been revised and extended over time as follows.

- MacFarling-Meure et al. (2006) extended the $CO_2$ and $CH_4$ records back through the last two millennia (green diamonds in Fig.s 2a and b), and increased sample density in the Industrial Period (green diamonds in Fig.s 3a and b). They also confirmed the LIA $CO_2/CH_4$ decrease as well as the 1940s $CO_2$ stabilisation, and produced a record of $N_2O$ (green diamonds

in Fig. 2d) which, in turn, overlaps with firn $N_2O$ measurements (green diamonds in Fig. 3d). The authors interpreted the $N_2O$ decrease of about 5 ppb during the LIA as additional evidence for the terrestrial origin of the LIA GHG decrease. The measurement uncertainty remained the same for $CO_2$ (1.2 ppm) as for Etheridge et al. (1996), but decreased slightly for $CH_4$ (from 5 ppb in Etheridge et al., 1998; to 4 ppb in MacFarling Meure et al., 2006). The uncertainty of the $N_2O$ measurements was 6.5 ppb. The authors also found an increase of $N_2O$ concentration of about 10 ppb between 675 and 800 CE, which does not seem to be related to any known climatic event.

- Rubino et al.(2013) revised the $\delta^{13}C$-$CO_2$ record (see yellow triangles in Fig.s 2c and 3c) by updating the calibration scale and revisiting the corrections applied in Francey et al. (1999) for blank, gravity and diffusion effects, using the revised CSIRO firn model (Trudinger et al., 2013) for the gravity and diffusion corrections (Trudinger et al., 1997). In doing so, they resolved the 0.2 ‰ discrepancy found between the Law Dome $\delta^{13}C$-$CO_2$ record and the South Pole $\delta^{13}C$-$CO_2$ firn record (the South Pole firn records have been reported in Fig 5, but see Rubino et al., 2013, for more details). They also increased sample density during the Industrial Period and applied the new chronology available for Law Dome ice (Plummer et al., 2012), which caused a shift of about 150 years for samples that are 2000 years old (see difference between the ages of green diamonds and yellow triangles in Fig. 2a). The age difference becomes negligible in the last millennium, as evident by comparing red squares and yellow triangles in Fig. 2c.

- Rubino et al. (2016) carried out additional $CO_2$ and $\delta^{13}C$-$CO_2$ measurements (see blue circles in Fig.s 2-3 a and c) from ice cores sampled at the Law Dome site of DSS0506 (Pedro et al., 2011). The data extended back to 1700 CE effective air age and provided additional evidence of consistent results between different ice cores and firn records where they overlapped. However, the increasing $CO_2$ trend measured in DSS0506 between 1700 and 1850 CE does not tightly match that previously attributed to recovery from the LIA (Etheridge et al., 1996).

It is also worth mentioning the results of two other studies performed using Law Dome ice and firn, which were sampled but not measured using CSIRO-GASLAB instruments.

- To investigate changes in Pre-Industrial sources of $CH_4$, Ferretti et al. (2005) produced a record of $\delta^{13}C$-$CH_4$ in Law Dome ice covering the last 2000 years (not shown). They reported unexpected changes of the global $CH_4$ budget, mainly attributed to variations of biomass burning emissions during the Late Pre-Industrial Holocene (LPIH) through an atmospheric box model (Lassey et al., 2000). The $\delta^{13}C$-$CH_4$ record from Ferretti et al. (2005) has not been included in ICEBASE because the air samples extracted in ICELAB were measured on a mass spectrometer not maintained by CSIRO-GASLAB. Therefore, the $\delta^{13}C$-$CH_4$ data are not on a CSIRO calibration scale and have not been included in ICEBASE.

- Park et al. (2012) measured oxygen and intramolecular nitrogen isotopic compositions of $N_2O$ (not shown) covering 1940 to 2005 in Law Dome firn air and archived air samples from Cape Grim (Tasmania). In doing so, they confirmed that the rise in atmospheric $N_2O$ levels is largely the result of an increased reliance on nitrogen-based fertilizers. These isotopic measurements are also not included in ICEBASE.

## 3.2 The new Law Dome GHG records and comparison with records from other sites.

Figures 4-5 shows the newly revised Law Dome GHG records (red circles). Following the rule-based selection described in the Supplementary Material, there are 299 ice core measurements for $CO_2$, 307 for $CH_4$ and 147 for $N_2O$ (compared to 212, 228 and 103 respectively in MacFarling Meure et al., 2006), and 86 for $\delta^{13}C$-$CO_2$ (compared to 58 in Francey et al., 1999; and 69 in Rubino et al., 2013). All of the major features described in previous publications are retained. However, the differences mentioned above can potentially influence the biogeochemical and climatic interpretation of these records. Given that the Law Dome GHG records are a major source of information for models used to predict the future behaviour of the Earth System (Graven et al., 2017; Köhler et al., 2017b; Meinshausen et al., 2017), in the following paragraphs we provide an explanation of the main reasons for these differences.

- Changes to the calibration scale result in small, mostly negligible, differences.

- All records (except the $\delta^{13}C$-$CO_2$) start in 154 CE (effective age for $CO_2$) rather than 0 CE. This causes a revision of air ages towards more recent times for all events recorded in the second last millennium (e.g. the 10 ppb increase of $N_2O$ between 675 and 800 CE discussed in MacFarling-Meure et al. (2006) is now dated 701-822 CE, Fig. 4d), but less than 2 year change in dating after about 1000 CE.

- Each data point has an uncertainty, which is independently calculated based on the weighting and flagging systems described in the Supplementary Material. The uncertainty does not include any additional uncertainty associated with inter-core variability. For example, based on comparisons between samples of the same ages, the discrepancy found between DSS0506 and other Law Dome cores in the period 1700-1850 CE suggests that the inter-core variability can potentially add a random, extra-uncertainty of up to 5 ppm. Further research is needed to precisely quantify the inter-core variability.

The following list compares the new Law Dome records with records from other sites and discusses the main differences:

- There is good agreement between the revised $CO_2/\delta^{13}C$-$CO_2$ Law Dome records and the $CO_2/\delta^{13}C$-$CO_2$ records from DML produced in ICELAB-GASLAB (blue triangles in Fig. 4-5 a and b). Once the different gas age distributions of the ice cores are taken into account, the two records are in very good agreement, with difference of less than 2 ppm for $CO_2$, and differences within error bars for $\delta^{13}C$-$CO_2$ (Rubino et al., 2016). Given that both records have been produced at CSIRO-ICELAB/GASLAB, we can exclude any possible inter-laboratory offset.

- There is also acceptable agreement (random differences up to 4 ppm) between the $CO_2$ Law Dome record and the $CO_2$ records from EDML and South Pole (Siegenthaler et al., 2005, white and green squares in Fig. 4b). Considering that the records have been produced in different laboratories (CSIRO-ICELAB/GASLAB and University of Bern), it is possible that part of the difference is explained by an inter-laboratory offset.

- There is a systematic difference (3 ppm on average) between the Law Dome $CO_2$ record and the WAIS $CO_2$ record in the Pre-Industrial (Ahn et al., 2012, see gray squares in Fig. 4b). Though small, the difference is of concern because it is systematic throughout the whole record. The two laboratories (CSIRO-ICELAB/GASLAB and Oregon State University) that produced these records have also run a comparison experiment to quantify the contribution of a possible inter-laboratory

offset to the total discrepancy (Ahn et al., 2012). The good agreement (measurements from the two laboratories were consistent within the 1σ analytical uncertainty) found by the inter-comparison experiment suggests that the discrepancy is mostly due to an inter-core difference.

- There is an increase in this difference to >8 ppm between the Law Dome $CO_2$ dip around 1610 CE and the WAIS $CO_2$ decrease during the LIA (Ahn et al., 2012, compare gray squares and red circles in Fig. 4b). WAIS is considered a high accumulation site and should retain the same events as those recorded at Law Dome. The difference is even more surprising when the tight agreement between the Law Dome $CH_4$ record and the WAIS $CH_4$ record (Mitchell et al., 2011) around this time is considered (compare red circles and grey squares in Fig. 4c). The consistency between the Law Dome and the WAIS $CH_4$ record rules out dating issues or large differences in smoothing of the atmospheric signals between the two sites. This suggests a chemical origin (in-situ production) of the $CO_2$ discrepancy, which is more likely to occur for $CO_2$ than for $CH_4$. This interpretation is supported by additional evidence of 6 ppm discrepancy (Köhler et al., 2017b) during the Last Glacial Maximum, Last Termination and Early Holocene between the EDC $CO_2$ record (Monnin et al., 2001, 2004) and the WAIS $CO_2$ record (Marcott et al., 2014).

- There is a difference of up to 0.15 ‰ (compare gray squares and red circles in Fig. 4a) between the Law Dome $\delta^{13}C\text{-}CO_2$ record (Rubino et al., 2013) and the WAIS $\delta^{13}C\text{-}CO_2$ record (Bauska et al., 2015). This difference is most likely due to an inter-laboratory offset, but may also indicate a contribution from the $CO_2$ discrepancy to its $\delta^{13}C$, or a combination of the two. It is not possible to quantify the inter-laboratory offset without running an inter-comparison study. However, it is possible to calculate a maximum effect of the *in-situ* $CO_2$ production on the $\delta^{13}C$ measured at WAIS, assuming that the 3 ppm extra-$CO_2$ measured in WAIS (compared to an average Pre-Industrial $CO_2$ concentration of 280 ppm measured at Law Dome) is totally due to the *in-situ* production, and that it all comes from carbonate-carbon ($\delta^{13}C = 0‰$), because organic-carbon with $\delta^{13}C = -27$ ‰, would make the WAIS $\delta^{13}C\text{-}CO_2$ more negative than the Law Dome $\delta^{13}C\text{-}CO_2$. Under this assumption, we calculate a possible shift of 0.07 ‰ through an isotope mass balance (=[0-(-6.55)]*3/283). This can only explain part of the discrepancy, but can go up to 0.14 ‰ if an extra amount of 6 ppm is assumed (Ahn et al., 2012; Köhler et al., 2017b). The Bauska et al. (2015) record agrees within uncertainties with the Francey et al. (1999) dataset. However, Rubino et al. (2013) is the only record to show consistency with all firn records and direct atmospheric measurements (see Fig.s 3c and 5a). This would suggest that the Rubino et al. (2013) is currently the most accurate record and should be used to set a pre-industrial baseline. However, no definite conclusion can be drawn until a thorough intercomparison study is carried out between the labs that have produced the WAIS and the Law Dome $\delta^{13}C\text{-}CO_2$ datasets (Oregon State University-University of Colorado-Institute of Arctic and Alpine Research, INSTAAR and CSIRO). It is important to resolve the difference between the Law Dome and the WAIS $\delta^{13}C\text{-}CO_2$ records in order to establish a Pre-Industrial baseline and, thus, a Pre-Industrial-to-Industrial $\delta^{13}C\text{-}CO_2$ difference. Setting a Pre-Industrial baseline could have consequences on the Last Glacial Maximum-to-Pre-Industrial $\delta^{13}C\text{-}CO_2$ difference as well (Schmitt et al., 2012). These values are useful for biogeochemical interpretation (Broecker and McGee, 2013; Krakauer et al., 2006).

- As expected, the Law Dome/WAIS $CH_4$ concentrations are lower than the NEEM high resolution $CH_4$ record (Rhodes et al., 2013, the white squares in Fig. 4c show the median $CH_4$ concentrations for 5 year time slices, after data outliers have been removed) by an amount which is consistent with an inter-hemispheric $CH_4$ difference of 40-60 ppb (Mitchell et al., 2013). Interestingly, the LIA $CH_4$ decrease measured at NEEM appears to start before the $CH_4$ decrease measured at Law Dome/WAIS. The age scale of the NEEM $CH_4$ record published in Rhodes et al. (2013) has been revised with the updated ice age scale published in Sigl et al. (2015) and the new estimate of $\Delta$age provided by Buizert et al. (Buizert et al., 2014). Mitchell et al. (2013) have synchronised the GISP2 $CH_4$ record with the WAIS $CH_4$ record to investigate changes of the Inter Polar Difference in the Pre-Industrial based on the reasoning that "the multidecadal events observed in both ice core records must have occurred simultaneously since the durations of the events were much larger than the atmospheric mixing time (~1 year)" (Mitchell et al., 2013). The NEEM $CH_4$ record has not been synchronised with the others, and there are multiple possible reasons, including age scale issues, different smoothing of the atmospheric signals at the different sites and inadequate sampling resolution, to explain the discrepancy found between the NEEM and the GISP2/Law Dome/WAIS $CH_4$ records during the LIA. A thorough investigation is out of the scope of this paper, but, in the future, this discrepancy should be resolved to obtain a precise synchronisation of all ice core records available over the LIA.

- The $N_2O$ records from Greenland (Flückiger et al., 1999, Eurocore and GRIP, gray and white squares in Fig. 4d, respectively) and from EDC (Flückiger et al., 2002, green squares in figure 4d) show higher scatter than the Law Dome $N_2O$ record. All records need higher sampling resolution to investigate changes of atmospheric $N_2O$ concentration over the last centuries with more confidence. A new $N_2O$ record from a high-resolution site is required to explore the real variations of $N_2O$ in the Pre-Industrial Period (Ryu et al., 2018).

### 3.3 The LIA and the 1610 CE CO₂ minimum in DSS (Law Dome).

In an attempt to produce $\delta^{13}C$ data around the Law Dome 1610 CE $CO_2$ minimum (inadequately sampled by Francey et al., 1999) and confirm the interpretation of its terrestrial origin (Rubino et al., 2016; Trudinger et al., 1999), in 2012 we measured 18 samples from DSS, the only core at Law Dome covering the whole LPIH. The results both for $CO_2$ and for $\delta^{13}C$ are significantly lower than the spline fit to the revised records from Law Dome (results not shown). On the contrary, the $CH_4$ concentration measured is very consistent with the spline fit to the revised $CH_4$ record. In the past, abnormally low $CO_2$ concentration was interpreted as a sign of post coring melting (PCM), $CO_2$ being much more soluble than $CH_4$. With PCM, the $N_2O$ concentration is usually low as well. However, in 7 of the 18 DSS samples that provided enough air to measure $N_2O$, the $N_2O$ concentration was, on average, not significantly lower than the spline fit to the revised Law Dome record. This argues against the hypothesis of PCM. Another possibility is the effect of clathrate formation which could alter $CO_2$ and $\delta^{13}C$, but issues due to clathrate can be ruled out because none of the ice cores in this study reached depths or temperatures sufficient for clathrate formation which can affect the extraction and measurement of enclosed gases and $^{13}C$-$CO_2$ in particular (e.g. Schaefer et al., 2011). We do not have a definite explanation for the low $CO_2$ and $\delta^{13}C$ measured, but the

results suggest that the DSS core may no longer be a reliable archive for $CO_2$. This experiment was conducted while we were measuring the DSS0506 $CO_2$ samples published in Rubino et al. (2016). During that survey, we also found a similar behaviour for 6 of the 34 DSS0506 samples measured. Since the two cores - DSS0506 and DSS - were stored in the same freezer in Hobart (Tasmania, Australia), with DSS0506 stored for a shorter period, we suggest the low $CO_2$ may be a recent
effect of storage (see Supplement to Rubino et al., 2016).

The LIA, and particularly the 1610 CE $CO_2$ event, is important for our understanding of the carbon cycle dynamics and the carbon-climate system in the past. It has been used to derive the $CO_2$ sensitivity to temperature (Cox and Jones, 2008; Rubino et al., 2016), it is the most prominent biogeochemical event during the LPIH, and it has even been suggested as the beginning of the new geologic era called the Anthropocene (Lewis and Maslin, 2015). Therefore, it is of fundamental
importance that we understand the amplitude of the minimum as recorded by the ice and the timing and likely size of the original atmospheric decrease before smoothing during firn diffusion and enclosure into bubbles.

## 3.4 Biogeochemical and climatic interpretation of the Law Dome GHG records

The Law Dome GHG records have been used for biogeochemical and climatic interpretation of changes in $CO_2$ (Joos et al., 1999; Joos and Bruno, 1998; Rubino et al., 2013; Trudinger et al., 2002a), $CH_4$ (Ferretti et al., 2005; Ghosh et al., 2015), and
$N_2O$ (Park et al., 2012) over the past decades to millennia. They are also used as reference atmospheric GHG records for model simulations of the carbon-climate system of the LPIH (Graven et al., 2017; Köhler et al., 2017b; Meinshausen et al., 2017). Here we present an overview of the insight obtained through interpretation of the Law Dome GHG records, and provide some perspective on the challenges we will have to face to obtain a deeper understanding of the carbon-climate system during the LPIH and the Industrial Period.
The biogeochemical interpretation of GHG variations depends on quantifying their sources and sinks. The concentration of GHGs in the atmosphere is the net result of the processes releasing GHGs to the atmosphere (sources) and processes removing GHGs from the atmosphere (sinks). The atmospheric circulation, then, redistributes GHGs assuming consistency in the reasonably well known patterns of air movement. Measuring the atmospheric concentration of GHGs provides one constraint on the net sources vs sinks. However, generally, multiple sources and sinks act simultaneously. Therefore,
multiple constraints are necessary to partition the contribution of each source or sink. Measuring the isotopic composition of each GHG provides an additional constraint (albeit usually with additional complexity), but there are also other independent constraints, such as the inter-hemispheric difference or evidence coming from other species, that help quantify the contribution of sources and sinks. Additionally, the net GHG emission strength is reflected in the rate of change of atmospheric concentration. So, ice core records that track rapid changes (i.e. high resolution) are best to infer budgets and
hence biogeochemical information before direct atmospheric observations. The Law Dome records provide the highest resolution among existing Antarctic ice cores. There have been recent studies looking into the effects of firn microstructure (including density layers) on bubble trapping (Burr et al., 2018; Fourteau et al., 2017; Gregory et al., 2014; Mitchell et al., 2015). Improved understanding of these processes, how they affect smoothing of atmospheric GHG signals, and their

incorporation into numerical models may lead to a more accurate quantification of the relationship between ice core GHG measurements at different sites and the original atmospheric variations.

$CO_2$: The two major reservoirs that can change atmospheric concentrations over years to millennia are the terrestrial biosphere (land) and the oceans. Fossil fuel and land use emissions have added to these over recent centuries. Measurements of $\delta^{13}C$-$CO_2$ have been used to quantify the contribution of land and ocean to the atmospheric $CO_2$ variations measured (Joos et al., 1999; Joos and Bruno, 1998; Trudinger et al., 2002a). For example, the interpretation of $CO_2$ and $\delta^{13}C$-$CO_2$ variations through a Double Deconvolution (Fig. 6a) has identified the terrestrial biosphere as the main contributor to the LIA $CO_2$ decline (Rubino et al., 2013, 2016; Trudinger et al., 2002a). This agrees with the findings of Bauska et al. (2015) who used the WAIS $CO_2$ and $\delta^{13}C$-$CO_2$ records to suggest that changes in terrestrial organic carbon stores best explain the observed multi-decadal variations in the $\delta^{13}C$-$CO_2$ and in $CO_2$ concentrations from 755 to 1850 CE. This agreement of interpretation from the DSS and WAIS records shows that what matters for the biogeochemical interpretation is the change in concentration over time, rather than the absolute concentration measured in different ice cores. The above studies assume consistency in the inter-hemispheric transport of Northern Hemisphere terrestrial emissions over multiple decades. Preliminary examination suggests Southern Hemisphere $\delta^{13}C$-$CO_2$ records over the last decades are more susceptible to multi-year changes in the ratio of eddy to mean advective interhemispheric transport (Francey and Frederiksen, 2016; Frederiksen and Francey, 2018) than is the case for $CO_2$ concentration, as a result of differences in isotopic equilibration that occur for the two transport modes.

An additional constraint for the biogeochemical interpretation of the LIA $CO_2$ decrease has recently come from the new interpretation (Rubino et al., 2016) of the record of carbonyl sulfide (COS, Aydin et al., 2008) from the SPRESSO ice core (South Pole Remote Earth Science and Seismological Observatory) and modelling of its increase over the LIA (Rubino et al., 2016). Rubino et al. (2016) showed that the simultaneous COS increase during the LIA confirms that the LIA $CO_2$ decline was caused by net terrestrial uptake due to cooling (heterotrophic respiration declining more than Gross Primary Production, due to its higher sensitivity on temperature changes), though a very recent paper estimating the amount of carbon taken up by land use change following the colonisation of the Americas by the Europeans (Koch et al., 2019) provides a different view. Nonetheless, the multi-species approach used by Rubino et al. (2016, e.g. using trends of $CO_2$, $\delta^{13}C$-$CO_2$ and COS) can provide multiple constraints to help understand the biogeochemical processes behind atmospheric $CO_2$ variations over the recent past.

The 1940s $CO_2$ plateau is a prominent feature in the Industrial part of the Law Dome $CO_2$ record, and occurs at a time of continued fossil fuel emissions. Taking into consideration the smoothing effects of firn diffusion and bubble trapping on the rate of change of potential atmospheric signals, an uptake of around 2-3 GtC/yr between 1942-49 would be required to explain the observed plateau. The Law Dome $\delta^{13}C$-$CO_2$ measurements suggest that the oceans were responsible for at least two thirds of this uptake (Rubino et al., 2013; Trudinger, 2000, section 6.4; Trudinger et al., 2002a). Bastos et al. (2016) used

the latest estimates of fossil fuel and land-use change emissions, ocean uptake reconstructions and terrestrial models, but was not able to explain the plateau, although they did not consider decadal variability in the ocean carbon sink that may have been important. Better understanding of the 1940s feature is needed to quantify how variable ocean and land carbon exchange can be on multi-year to decadal timescales, and how this variability relates to climate variability. Improved

understanding is expected to come from more high-precision, high-time-resolution ice core measurements to confirm $\delta^{13}C$-$CO_2$ variation through the Industrial Period, additional constraints such as COS, better understanding of the smoothing effects on trapped air along with improved modelling of other influences on atmospheric $\delta^{13}C$-$CO_2$ such as the effect of climate on isotopic discrimination (Randerson et al., 2002; Scholze et al., 2003).

CH$_4$: There are multiple natural sources/sinks: geological, wetlands, wildfires, termites, and ocean sediments are the main

sources, while oxidation by tropospheric species [OH], oxidation by stratospheric species [OH, Cl and O(1D)], oxidation in soils and reactive chlorine in the marine boundary layer are the main sinks. Also for $CH_4$, measurements of its isotopic composition ($\delta^{13}C$-$CH_4$ and $\delta D$-$CH_4$) have helped quantify the contribution of land vs ocean to the measured atmospheric $CH_4$ variations (Ferretti et al., 2005; Mischler et al., 2009; Sapart et al., 2012). However, because there are more distinct $CH_4$ source types than isotopic tracers, and more spatially distributed sources than can be resolved by the geographically

restricted suite of ice cores (despite the fact that the ice cores of Antarctica and Greenland are both known to provide reliable $CH_4$ records), a unique solution for the history of $CH_4$ sources is not possible. Nonetheless, the full suite of isotopic tracers and bipolar ice core data provides important boundary conditions for testing hypothetical $CH_4$ source/sink histories, allowing elimination of large classes of scenarios. By measuring $\delta^{13}C$-$CH_4$ in Law Dome ice, Ferretti et al. (2005) provided evidence of a remarkable decrease of pyrogenic $CH_4$ during the last millennium (Fig. 6b). This interpretation was confirmed by

Mischler et al. (2009) and Sapart et al. (2012), who measured both $\delta^{13}C$-$CH_4$ and $\delta D$-$CH_4$ in ice cores from Antarctica and Greenland respectively. They also found an increasing agricultural source of $CH_4$ throughout the last millennium, with most of the change between the 1500s and the 1600s, supporting the hypothesis of a pre-industrial anthropogenic influence on atmospheric $CH_4$. Additional source information is provided by measurements of carbon monoxide (CO) and its isotopes ($\delta^{13}C$-CO and $\delta^{18}O$-CO, Wang et al., 2010), through evidence of variations of biomass burning, and $^{14}CH_4$, which identifies

$CH_4$ emissions from fossil sources. However, the measurement of $^{14}CH_4$is limited so far to large air samples extracted from firn (Lassey et al., 2007b, 2007a) or from large ice samples collected where glacial-age ice outcrops at the surface (Petrenko et al., 2009, 2017).

N$_2$O: There are also multiple natural sources and sinks of N$_2$O, both on land and in the ocean. The main sources are microbiological processes, especially in tropical soils, while the main sinks are photochemical reactions in the atmosphere.

To the best of our knowledge, there are only two attempts to use $\delta^{15}N$-N$_2$O and $\delta^{18}O$-N$_2$O records to better constrain the land vs ocean sources of N$_2$O over the last century (Park et al., 2012; Prokopiou et al., 2017), and these used the larger volumes of air available in firn. No study has extended the investigation to the last millennia, but there is an analysis covering the Last Deglaciation by Schilt et al.(2014). As already mentioned, the Pre-Industrial inter-hemispheric N$_2$O difference is also poorly

constrained. Thus, there is room for vast improvement to understand the N cycle from measurements of $N_2O$ concentration and its isotopes in ice cores (Schilt et al., 2014). However, there is a risk of *in-situ* production of $N_2O$, especially in Greenland ice (Flückiger et al., 2002). More records, with high sampling resolution, from both hemispheres are needed to confirm the features found in Law Dome (MacFarling Meure et al., 2006) and understand the causes of the changes in $N_2O$
concentration over time (Ryu et al., 2018).

The climatic interpretation of the Law Dome GHG records has generally been carried out by comparing the timing of GHG variations and temperature changes, and testing hypotheses of mechanistic relationships between the two with coupled carbon cycle-climate models.

Medieval Climate Anomaly: An important climatic event of the LPIH is the Medieval Climate Anomaly (MCA, roughly
950–1250 CE), which showed higher temperature in some regions (Goosse et al., 2005; Mann et al., 2009). During the MCA, there are generally higher levels of GHGs, but the timings of increase vary from one gas to another, with $N_2O$ showing a rise between 701 and 822 CE (well before the start of the MCA), $CO_2$ increasing between 950 and 1200 CE, and $CH_4$ showing some variability superimposed on a long-term increasing trend (Fig. 4b-d).

Little Ice Age: The main climatic event of the last two millennia is the LIA (roughly 1400-1700 CE, Mann et al., 2008;
Neukom et al., 2014; Pages2k, 2013). There is a clear decrease of all GHG during the LIA, which, together with the other evidence mentioned above, suggests that all processes releasing GHG from land slowed down during the cold phase (Fig. 4 b-d). However, as for the MCA, the change in concentration is not simultaneous for all GHG. While both $CO_2$ and $N_2O$ seem to decrease starting around 1550 CE (but $N_2O$ would need higher sampling resolution to confirm this), $CH_4$ has a later decline, starting around 1580 CE. Also, the $CH_4$ decrease seems to last for a shorter period of time, ending around 1610 CE,
whereas the $CO_2$ low is maintained longer, ending about 1750 CE. At the same time, there was a significant decrease of biomass burning (Ferretti et al., 2005; Mischler et al., 2009; Sapart et al., 2012; Wang et al., 2010), interpreted to be mostly a consequence of decreased fire emissions. While the relationship between $CO_2$ and temperature variation has been used to infer the sensitivity of $CO_2$ to temperature (Cox and Jones, 2008; Rubino et al., 2016), confirming and quantifying a positive feedback of terrestrial carbon with temperature (Rubino et al., 2016), it is now time to investigate the regional contribution to
the total $CO_2$ change, as attempted by Bauska et al. (2015), and the contribution from different processes within the terrestrial biosphere, such as net primary production, heterotrophic respiration and biomass burning. There are regional (Mann et al., 2008), continental (Pages2k, 2013) and hemispheric (Neukom et al., 2014) temperature reconstructions that can be used to drive models describing the relationship between climate and carbon cycle to quantify the contribution from each region to the total $CO_2$ decrease (Fig. 6c). There are also hydro-climatic reconstructions (Cook et al., 2009, 2010, 2015)
providing evidence of dry and wet periods over the LPIH (Fig. 6d), which can be used, together with records of charcoal (Marlon et al., 2013, 2016) and biomass burning tracers in ice cores (Grieman et al., 2017) to quantify the contribution of declining biomass burning in each region to the total $CO_2$ decrease. The emissions from anthropogenic land use change have also been quantified for each world region (Pongratz and Caldeira, 2012, see Fig. 6e), and can be used to subtract the human contribution from the total $CO_2$ change, even though there is a debate on the amount of land use change following the

European colonisation of the Americas (Koch et al., 2019). We suggest that the LIA provides a suitable epoch to further study carbon cycle-climate feedbacks, for predictions of the future carbon-climate system, particularly to understand the role of different regions of the world on changes of atmospheric chemistry and biogeochemical fluxes and carbon pools. The consequences of the LIA climatic changes on societal development are important for understanding why different communities were more or less vulnerable, resilient or even adaptive (Degroot, 2018). Being able to quantify which regions have been more vulnerable to past climate change, also in terms of the response of the natural carbon cycle, could help plan future adaptation strategies.

## 4 Conclusions

The records of GHG ($CO_2$, $CH_4$, $N_2O$) concentrations and the isotopic composition of $CO_2$ ($\delta^{13}$C-$CO_2$) from the Law Dome ice cores are among the most important sources of information for models trying to predict the future behaviour of biogeochemical cycles and their influence on the climate system. The records of $CO_2$, $CH_4$, $N_2O$ and $\delta^{13}$C-$CO_2$ are constantly being updated with new measurements, and revised for changes in calibration scales and corrections for the effects of laboratory extractions and those of gravity and diffusion in firn. This paper has provided an in-depth explanation of the procedures used to extract and measure the samples in CSIRO-ICELAB, and store, correct and select the results obtained. Given the widespread use of the datasets produced at CSIRO ICELAB-GASLAB, it is important to provide a track record of the reasons for changes carried out over time, and keep the records open to the scientific community. This should help with their use in modelling and avoid misinterpretation (for example, the mistake spotted by Köhler et al., 2017a, in the paper from Machado and Froehner, 2017).

The GHG records from Law Dome have already provided significant insight into biogeochemical and climatic feature over the last centuries. However, there is room for deeper understanding, for example studying the influence of different regions on variations in atmospheric chemistry. Also, there are unresolved discrepancies (e.g. the LIA $CO_2$ decrease) which need to be resolved and Law Dome appears to provide the most suitable ice cores for high-resolution investigation of atmospheric changes over the last centuries. Considering that:

- All GHG records in ice cores are a smoothed representation of the real atmospheric history;
- DSS is the highest accumulation rate site ever sampled in Antarctica recording the LIA $CO_2$ event;
- There is the risk that the WAIS core is affected by *in-situ* production of $CO_2$;
- Accurate $CO_2$ record have not been derived from Greenland ice cores

we suggest that there is a need to sample a new, clean and deep ice core from Law Dome, to confirm or improve our knowledge of the atmospheric LIA $CO_2$ decrease and other rapid changes in atmospheric composition during Pre-Industrial millennia.

Finally, there are open questions about the real size of past atmospheric variations of some species, such as $N_2O$ and COS, and the reasons for those variations, which, once resolved with new measurements, could provide additional understanding and insights, useful to better predict the future behaviour of biogeochemical cycles and their influence on the climate system.

## Author contribution

MR conceived the database structure, performed measurements between 2009-2013 and led writing of the paper. DME supervised the process, performed measurements and collected samples. RH wrote the database queries and procedures in Microsoft SQL server and developed the web-based user interface with Microsoft Visual Studio. DPT, CEA, RJF, RLL, LPS and DAS carried out measurements of samples and standards in GASLAB, and performed calibration scale updates. CMT carried out the numerical modelling, including the CSIRO firn model, the double deconvolution and the Spline fit to the data. MAJC, TDVO and AMS collected, stored and distributed samples and contributed to ice dating.

## Acknowledgments

We acknowledge the long-term support provided to this work by CSIRO, the Department of Energy and Environment, the Australian Antarctic Division and Australian Antarctic Program, and the many people involved in collecting the several ice cores and firn air smaples from Law Dome since the 1980s. In particular chief driller of the DSS ice core Alan Elcheikh (1960-2019). We thank Rachael Rhodes and Michael Sigl for providing updated ice and gas age scales for NEEM. We also thank to the 2 anonymous referees for their comments, which have helped improving the original manuscript.

## Data Availability

Data connected with this paper are available in the CSIRO Data Access Portal (https://doi.org/10.25919/5bfe29ff807fb). Each species has been ordered by core and by age. In detail, for each species and each record the following fields are available (Rubino et al., 2018):

- Sample ID
- Ice Age
- Gas age
- Value
- Uncertainty

For each species, the calculated spline fit with time steps of 1 year and growth rate are also given. The spline fits attenuate variations with periods of less than 20 years by 50% for $CO_2$ and $CH_4$, 100 years for $N_2O$ and 50 years for $\delta^{13}CO_2$. When

using these data please consider citing the original publications from which the data underlying this compilation have been taken

.

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

Figure captions

Figure 1: Map of the Law Dome region, slightly modified from Smith et al. (2000), showing the location of the drilling sites DE08, DE08-2, DSS, DSS0506 and DSSW20K discussed in the text. Dotted lines are accumulation isopleths (kg m$^{-2}$ yr$^{-1}$) and unbroken lines are elevation contours (meters above sea level). The inset shows the location of the region in Antarctica

Figure 2: Published Pre-Industrial Period (1-1900 AD) GHG records from Law Dome ice extracted and measured at CSIRO ICELAB-GASLAB. (a) $CO_2$; (b) $CH_4$; (c) $\delta^{13}C$-$CO_2$ and (d) $N_2O$

Figure 3: Published and unpublished Industrial Period (1750-2000) GHG records from Law Dome ice and firn, extracted and measured at CSIRO ICELAB-GASLAB. (a) $CO_2$; (b) $CH_4$; (c) $\delta^{13}C$-$CO_2$ and (d) $N_2O$

Figure 4: Revised records (100-1900 AD) of (a) $\delta^{13}C$-$CO_2$; (b) $CO_2$; (c) $CH_4$ and (d) $N_2O$ from Law Dome ice compared to published records from other sites: WAIS-$\delta^{13}C$-$CO_2$ from Bauska et al. (2015), WAIS-$CO_2$ from Ahn et al. (2012), DML-$CO_2$ and -$\delta^{13}C$-$CO_2$ from Rubino et al. (2016), EDML- and South Pole-$CO_2$ from Siegenthaler et al. (2005), WAIS-$CH_4$ from Mitchell et al. (2011), NEEM-CH4 from Rhodes et al. (2013) and GISP2-$CH_4$ from Mitchell et al. (2013), Eurocore- and GRIP-$N_2O$ from Flückiger et al. (1999), EDC-$N_2O$ from Flückiger et al. (2002).

Figure 5: Revised records (1750-2010 AD) of (a) $\delta^{13}C$-$CO_2$; (b) $CO_2$; (c) $CH_4$ and (d) $N_2O$ from Law Dome ice and firn compared to the South Pole firn records of $\delta^{13}C$-$CO_2$, $CO_2$, $N_2O$ and to published records from other sites: WAIS-$\delta^{13}C$-$CO_2$ from Bauska et al. (2015), WAIS-$CO_2$ from Ahn et al. (2012), DML-$CO_2$ and -$\delta^{13}C$-$CO_2$ from Rubino et al. (2016), EDML- and South Pole-$CO_2$ from Siegenthaler et al. (2005), WAIS-$CH_4$ from Mitchell et al. (2011), NEEM-CH4 from Rhodes et al. (2013) and GISP2-$CH_4$ from Mitchell et al. (2013), Eurocore-$N_2O$ from Flückiger et al. (1999).

Figure 6: Biogeochemical and climatic interpretation of the Law Dome GHG records: (a) atmospheric $CO_2$ fluxes from (negative values on the y axis) and to (positive values on the y axis) the terrestrial biosphere (land: green line) and the ocean (blue line), resulting from the Double Deconvolution of $CO_2$ and $\delta^{13}C$-$CO_2$ (Rubino et al., 2016). (b) Flux of atmospheric $CH_4$ from biomass burning (Ferretti et al., 2005); (c) Temperature variations of different continents in the Northern Hemisphere (Asia: yellow line, Europe: blue line; North America: red line) from Pages2k (2013); (d) Palmer Drought Severity Index (PSDI) of different continents in the Northern Hemisphere (Asia: yellow line, Europe: blue line; North America: red line) from Cook et al. (2009, 2010, 2015) (d) Atmospheric $CO_2$ fluxes from different continents in the Northern Hemisphere (Asia: yellow line, Europe: blue line; North America: red line) due to Pre-Industrial land use change from Pongratz et al. (2012).

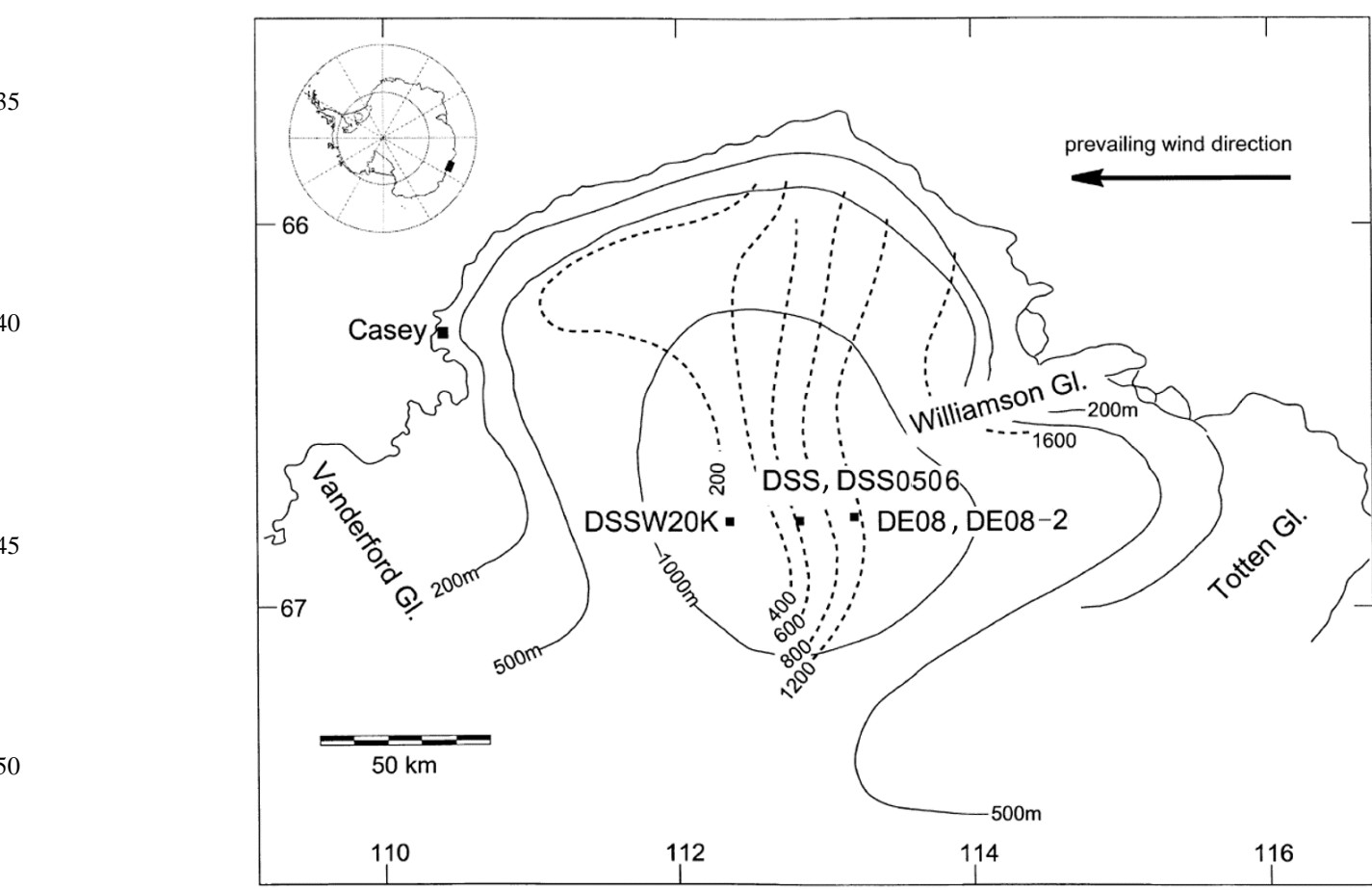

Figure 1: Map of the Law Dome region, slightly modified from Smith et al. (2000), showing the location of the drilling sites DE08, DE08-2, DSS, DSS0506 and DSSW20K discussed in the text. Dotted lines are accumulation isopleths (kg m$^{-2}$ yr$^{-1}$) and unbroken lines are elevation contours (meters above sea level). The inset shows the location of the region in Antarctica

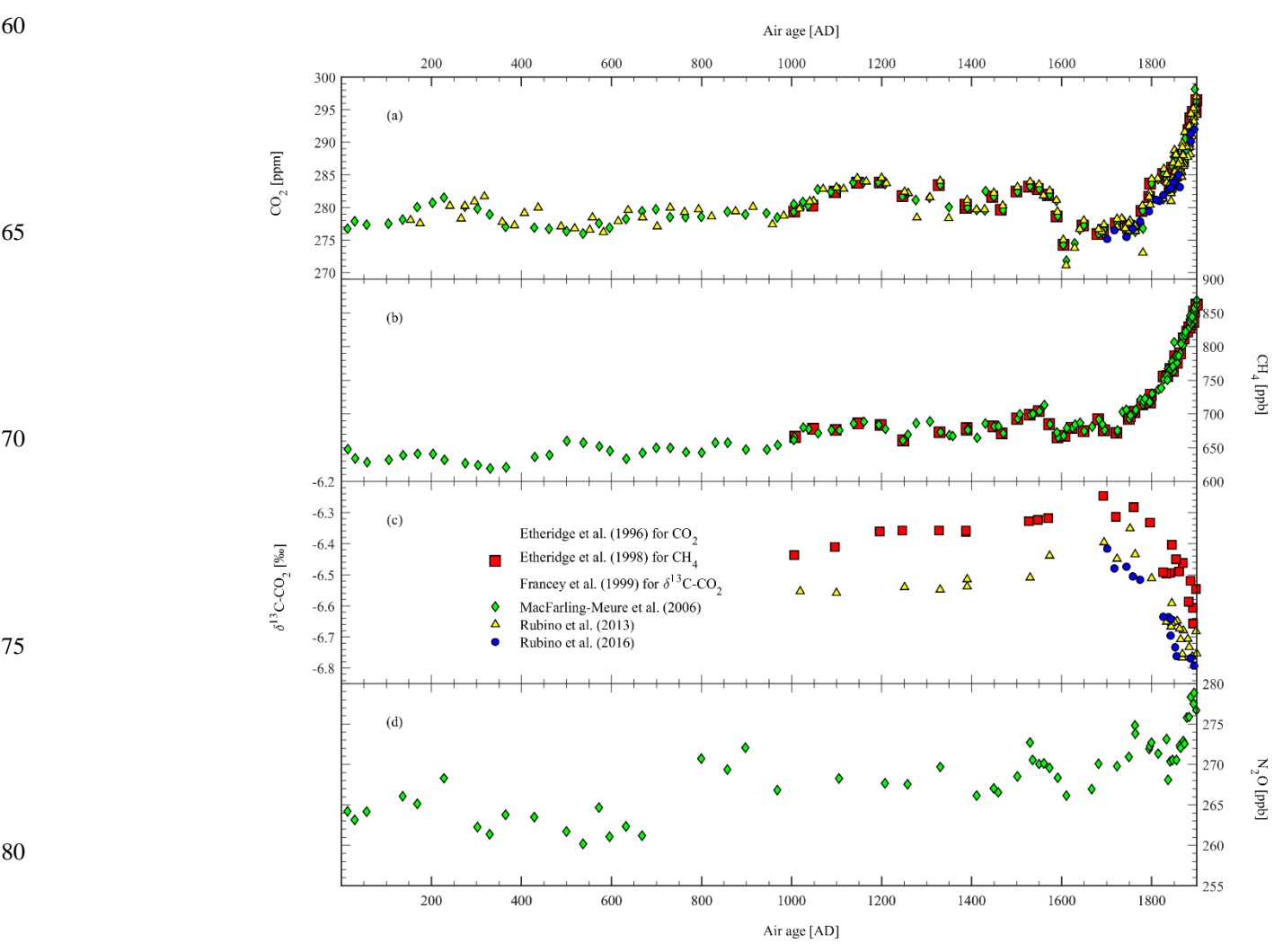

Figure 2: Published Pre-Industrial Period (1-1900 AD) GHG records from Law Dome ice extracted and measured at CSIRO ICELAB-GASLAB. (a) $CO_2$; (b) $CH_4$; (c) $\delta^{13}C\text{-}CO_2$ and (d) $N_2O$

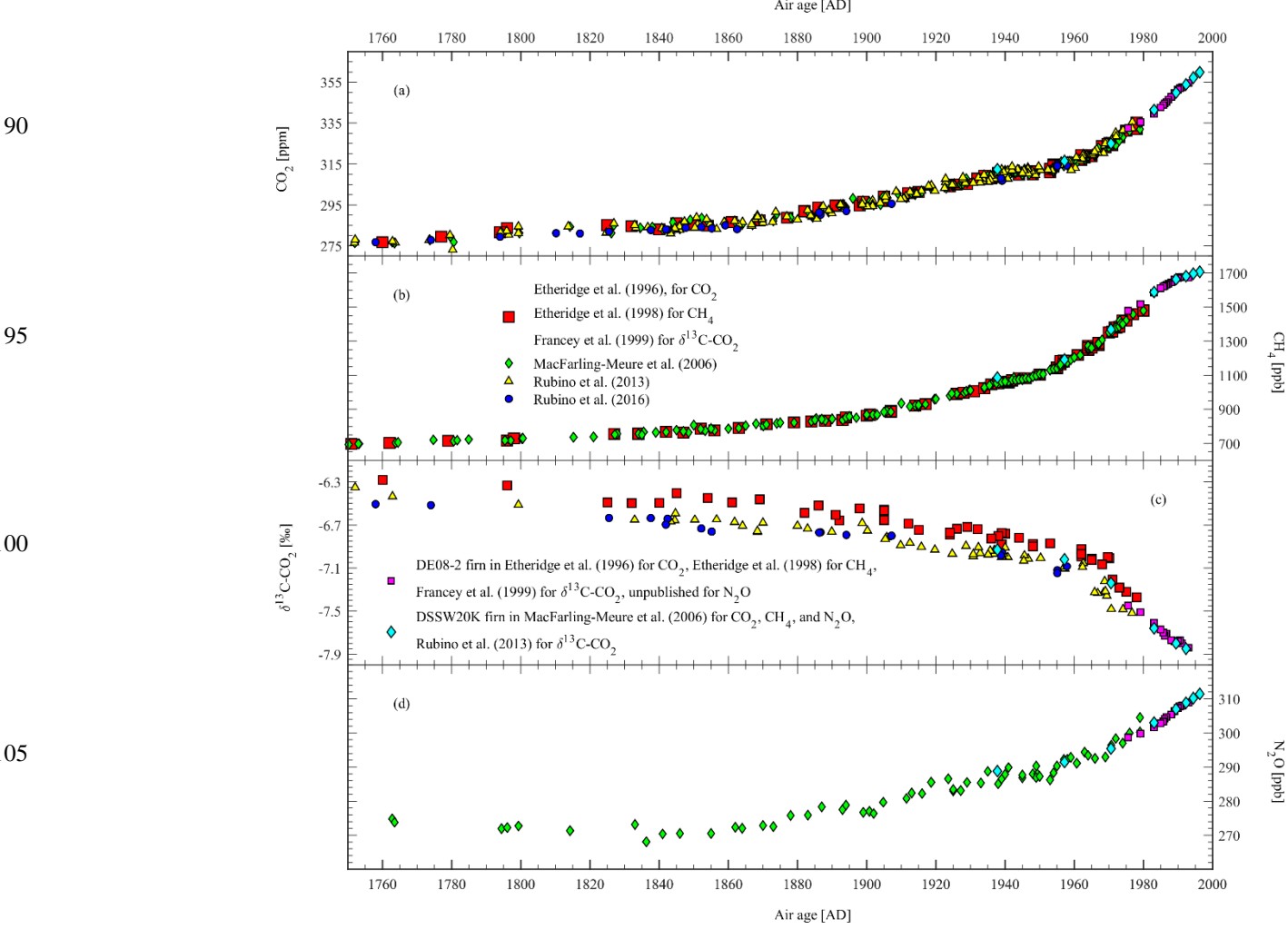

Figure 3: Published and unpublished Industrial Period (1750-2000 AD) GHG records from Law Dome ice and firn, extracted and measured at CSIRO ICELAB-GASLAB. (a) $CO_2$; (b) $CH_4$; (c) $\delta^{13}C$-$CO_2$ and (d) $N_2O$

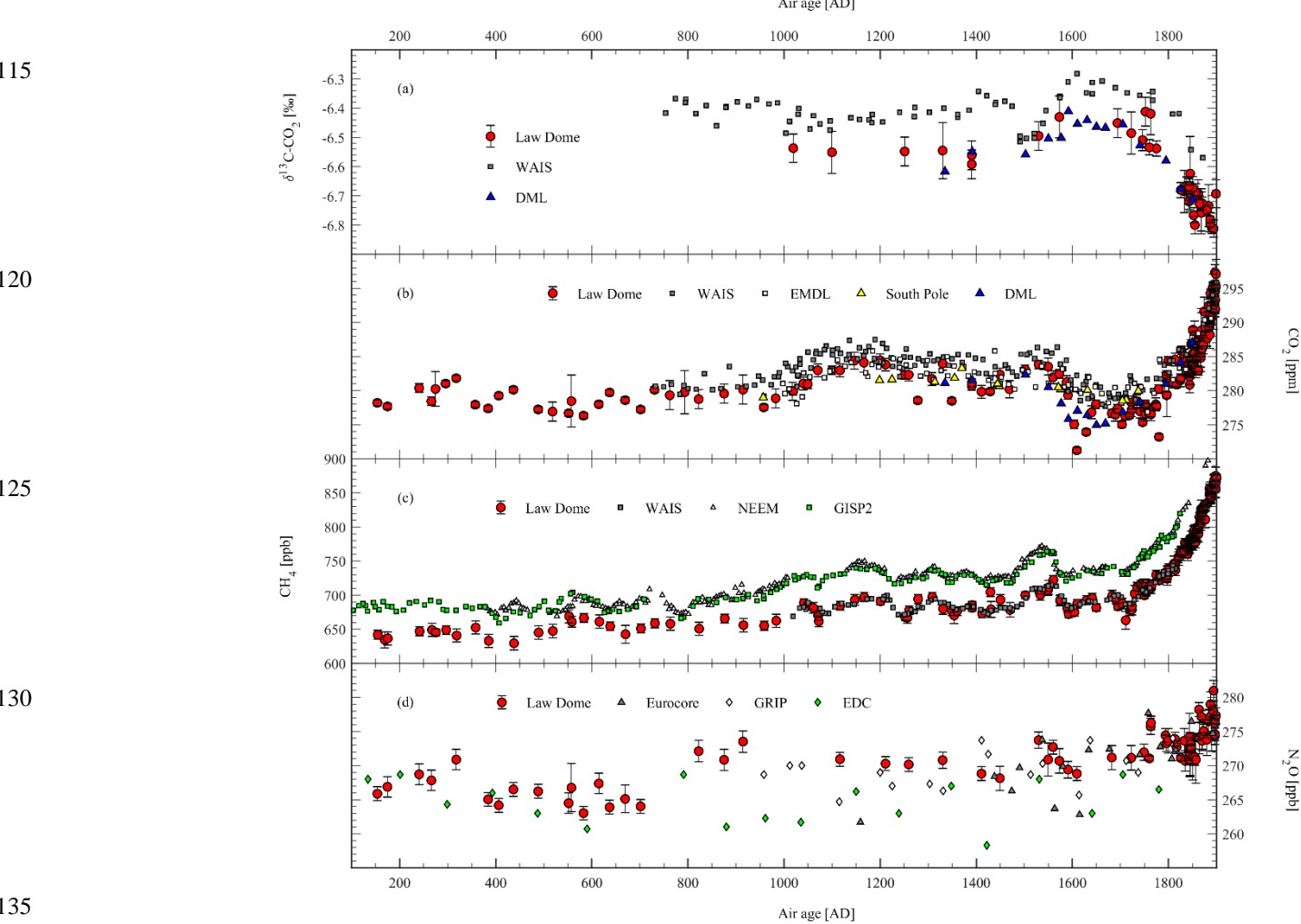

Figure 4: Revised records (100-1900 AD) of (a) $\delta^{13}$C-CO$_2$; (b) CO$_2$; (c) CH$_4$ and (d) N$_2$O from Law Dome ice compared to published records from other sites: WAIS-$\delta^{13}$C-CO$_2$ from Bauska et al. (2015), WAIS-CO$_2$ from Ahn et al. (2012), DML-CO$_2$ and -$\delta^{13}$C-CO$_2$ from Rubino et al. (2016), EDML- and South Pole-CO$_2$ from Siegenthaler et al. (2005), WAIS-CH$_4$ from Mitchell et al. (2011), NEEM-CH4 from Rhodes et al.

140     (2013) and GISP2-$CH_4$ from Mitchell et al. (2013), Eurocore- and GRIP-$N_2O$ from Flückiger et al. (1999), EDC-N2O from Flückiger et al. (2002).

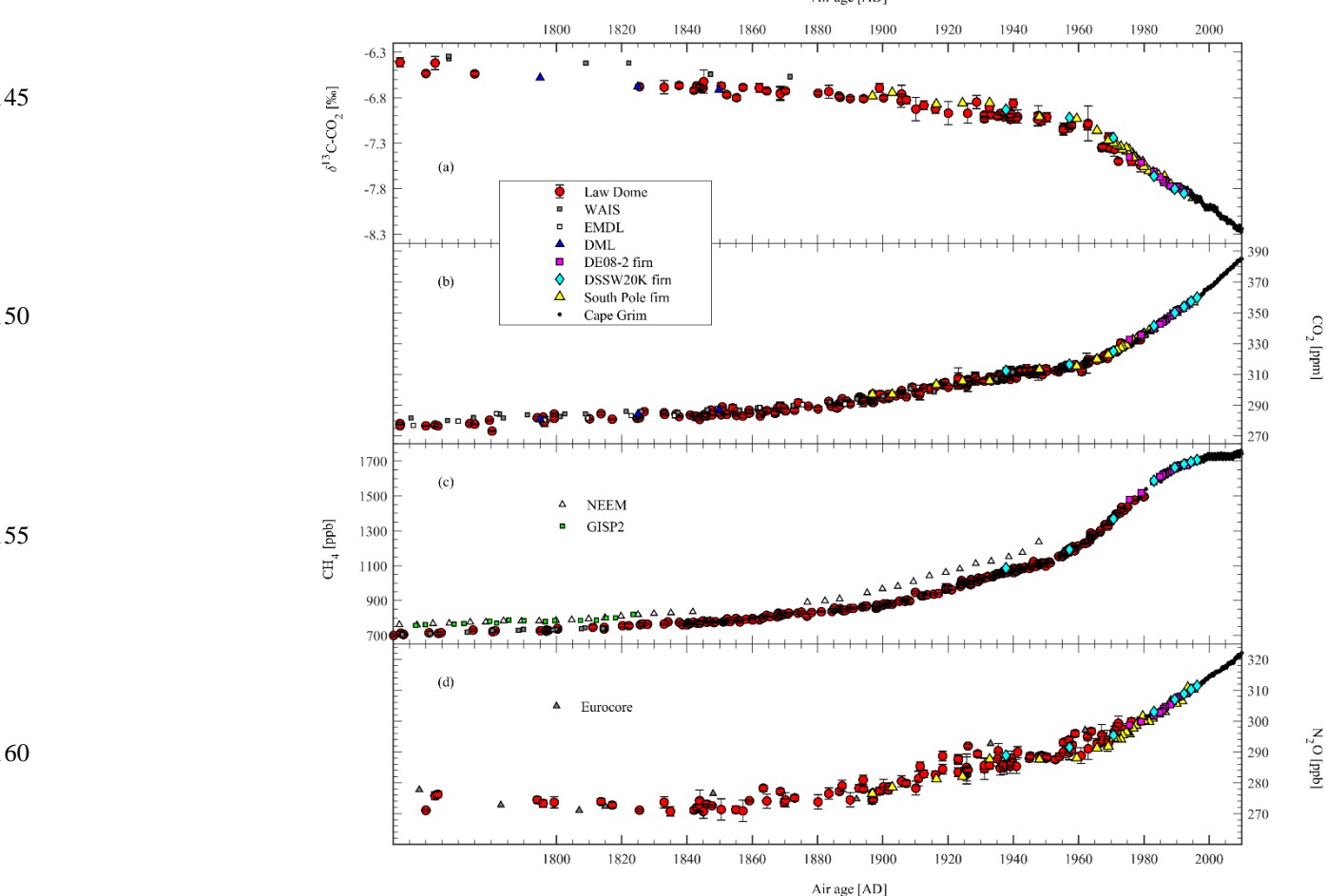

Figure 5: Revised records (1750-2010 AD) of (a) $\delta^{13}$C-$CO_2$; (b) $CO_2$; (c) $CH_4$ and (d) $N_2O$ from Law Dome ice and firn compared to the South Pole firn records of $\delta^{13}$C-$CO_2$, $CO_2$, $N_2O$ and to published records from other sites: WAIS-$\delta^{13}$C-$CO_2$ from Bauska et al. (2015), WAIS-$CO_2$ from Ahn et al. (2012), DML-$CO_2$ and -$\delta^{13}$C-$CO_2$ from Rubino et al. (2016), EDML- and South Pole-$CO_2$ from Siegenthaler et al. (2005), WAIS-$CH_4$ from Mitchell et al. (2011), NEEM-CH4 from Rhodes et al. (2013) and GISP2-$CH_4$ from Mitchell et al. (2013), Eurocore-$N_2O$ from Flückiger et al. (1999).

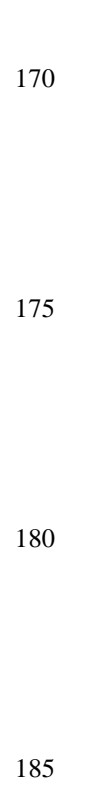
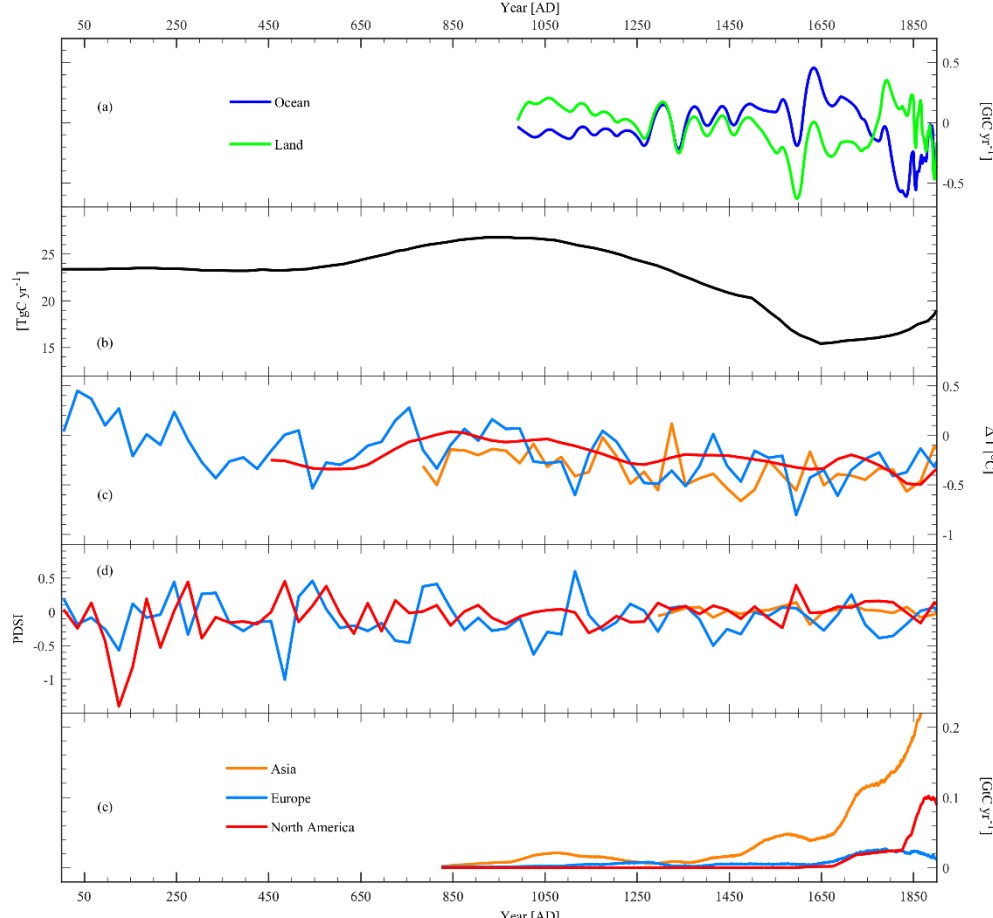

Figure 6: Biogeochemical and climatic interpretation of the Law Dome GHG records: (a) atmospheric $CO_2$ fluxes from (negative values on the y axis) and to (positive values on the y axis) the terrestrial biosphere (land: green line) and the ocean (blue line), resulting from the Double Deconvolution of $CO_2$ and $\delta^{13}C$-$CO_2$ (Rubino et al., 2016). (b) Flux of atmospheric $CH_4$ from biomass burning (Ferretti et al., 2005); (c) Temperature variations of different continents in the Northern Hemisphere (Asia: yellow line, Europe: blue line; North America: red line) from Pages2k (2013); (d) Palmer Drought Severity Index (PSDI) of different continents in the Northern Hemisphere (Asia: yellow line, Europe: blue line; North America: red line) from Cook et al.(2009, 2010, 2015) (e) Atmospheric $CO_2$ fluxes from different continents in the Northern Hemisphere (Asia: yellow line, Europe: blue line; North America: red line) due to Pre-Industrial land use change from Pongratz et al. (2012).