# Peer review of "S1 ICELAB database"

_Earth System Science Data, 2018_

## Referee Comment (RC1) · Anonymous Referee #1 · 24 Jan 2019

The paper "Revised records of atmospheric trace gases... from Law Dome, Antarctica" by Rubino et al. represents an important contribution to the subject of historical radiative forcing changes of the three major greenhouse gases CO2, CH4 and N2O. Moreover, these data and the additional d13CO2 information are most instrumental to improve our understanding of the natural and anthropogenic changes of the biogeochemical cycles of these gases.

The ICELAB greenhouse gas concentration data are amongst the most precise ice core data available for the last about 2000 years and also the d13CO2 data, despite some corrections and data screening required, are of sufficient quality and essential

for our understanding of carbon cycle changes. Due to the continuing progress in ice core analyses and the understanding of processes affecting the ice core gas archive, the Law Dome data have experienced several updates and corrections, which cannot be assessed or fully comprehended by the non-specialist. As the Law Dome data represent such an important asset in the climate change debate and are heavily used by climate modelers, it is important that an up-to-date data set is available online and a reference document describing this data set and the corrections exists. Accordingly, I strongly support publication of the paper and the connected data set by Rubino et al. in ESSD.

Having said that, the paper obviously does not go deep into the interpretation of the data sets and essentially reviews what has been said in previous papers before. As a reference document for the data set, I think this is mostly o.k. However, to completely fulfill the expectations for such a reference document, I missed some essential information that should be added in a revised version of the manuscript. These issues and some minor language correction are discussed in detail in the annotated version of the combined main text and supplement attached to this review.

The most important issues are:

- the issue of offsets in the d13CO2 data with previous versions of the data set and with data sets from other work is not sufficiently discussed yet. The corrections performed on earlier versions of the Law Dome data set are not sufficiently described in the text to allow the reader to make a quality assessment. In my opinion, it is not enough in this case to just cite older papers.

- it is a pity that the CH4 and N2O isotopic data from Law Dome measured in other labs is not included in the data base. Although these data sets cannot be continuously updated in a way the CSIRO data can, they should at least be included in the data base as published.

- in the supplement the way how ages and corrections are derived (using firn modeling?) is not sufficently discussed. This needs significantly more detail to assess the quality of the data.

Please also note the supplement to this comment:
https://www.earth-syst-sci-data-discuss.net/essd-2018-146/essd-2018-146-RC1-supplement.pdf
* * *

---

## Referee Comment (RC2) · Anonymous Referee #2 · 25 Jan 2019

This paper by Rubino et al. compiles revised GHG records (CO2, CH4, N2O, d13CO2) from Law Dome climatic archives. Law Dome is located in East Antarctica and is a unique high accumulation site which allows to study in details the recent atmospheric history. Revising and synthesizing gas data from Law Dome ice cores is thus a really valuable effort. This paper can certainly contribute to enhance the use of such data. For these reasons, the paper should be published.

The paper mostly includes a review of previous interpretations, as a base to identify where future Law Dome studies should focus, and where more lab intercomparison studies should be conducted. These advice for future work are nicely identified. In-

terpretations reported in the manuscript where published earlier, thus I focus here on minor comments and technical comments (related to Supplement). I note that the paper is well written and was easy to read.

Specific comments :

P2, line 15- 16 : I suggest to rephrase "it is extremely difficult to separate the impacts of anthropogenic increases in CO2 on carbon sinks from the impacts of global warming or increased CO2 concentration on these sinks.". This statement was unclear for me.

P3, line 7: Rhodes et al., 2016 (Climate of the Past) include more CH4 data from Greenland.

P5 sect. 2.1: use the same unit for all accumulation data, for homogeneity.

P6 line 23: typo "I think".

P11 Line 20: I understand that the atmospheric mixing between Northern hemisphere and Southern hemisphere is fast enough so [CH4] would exhibit almost simultaneous trend in both hemispheres. Here the shift in in LIA CH4 decrease seems to be about 40 yrs (Fig. 3). Similar shift seems to exist at the onset of the industrial period CH4 increase. Can we explain such shift with Age Scale uncertainty? Maybe discuss this shift by providing more quantitative estimation.

Figures :

I find the figures difficult to read: I would advise to increase size for labels and titles.

Figure 4 : this figure highlight two important past findings (Rubino et al., 2016 ; Ferreti et al., 2005). It also includes data that can potentially be relevant for further studies and interpretations. I am not sure these data (b, d, and e) need to be plotted; likely description in the manuscript is enough. If the authors want to keep these data as part of plot 4, I would recommend to clarify the figure, e.g. the panel b and c can be shifted, so the figure reports first ice core data, and second complementary climatic data.

[Figure]

Technical comments related to Supplement:

My main concern is about the way uncertainties are calculated by multiplying the blank uncertainty with a factor u.f. . I find this process complex, not fully understandable for data users, and maybe operator-dependent.

- To me, the blank uncertainty is independent from other sources of uncertainties (e.g., dispersion of results observed for replicated measurements on of the same sample). Independent uncertainties when propagated do not multiply each others.

- We can observe here that for a sample where u.f. = 1 (i.e., qf and mq = fair or good), the data uncertainty is reduced to only the blank uncertainty, ignoring for example that different replicated measurements of the same sample will likely not be exactly all the same.

- The u.f. factor includes many parameters (flags, and criteria associated to weights), and some of them are not related to uncertainty. As an example, the first parameter is "melt layer". A melt layer can, e.g., results in high methane concentration (due to in situ production), but the measurement uncertainty of such high concentration should not be different from a regular sample. Just higher concentration will be measured. A melt layer sample does not have the quality required for reconstructing past CH4, but its measurement could be of great quality! Overall, I would advise that the authors identify more clearly what causes uncertainties, instead of considering everything.

- Some of the flags and criteria associated to weights seem subjective, and maybe operator-dependent in their evaluation (at least this is what I feel when reading the Supplement).

- When all criteria reports "reject" (i.e., u.f. = 4), the data is not rejected, but the uncertainty is increased more. It seems to me that these data should be excluded (as suggested by the wording "fatal problem").

- It would be great, if this approach is kept, to provide the full list of flags and weights

[Figure]

Others technical comments:

- What are the typical blanks observed, and typical blank uncertainties observed?

- I am not convinced that CO concentration is a good tool to evaluate the quality of a measurement (similarly to "melt layer", see before), or the quality of a sample. CO can be produced by chemical processes (the authors mention biological production of CO, citation is missing for that), but to my knowledge no collocated productions of CO and, e.g. $CO_2$, $CH_4$ or $N_2O$ have been reported so far in ice cores. The processes involved could be different, and a sample compromised for CO can be of good quality for others analyses. Ambient CO is often higher than what is in ice core bubbles, but this is also clear with $CH_4$.

---

## Author Comment (AC1) · 8 Mar 2019

**Authors' comment**

Before providing our answers to the referees' comments, we need to make the Editor and the referees aware of a change in the manuscript we have decided to introduce because of a new article, which has just been published in Quaternary Science Reviews (Koch et al., 2019). We felt that it is important to mention the finding of that article in our paper because of the possible consequences on the interpretation of the greenhouse gas records. Therefore, we have added a sentence at page 15 line 13-16: "*Rubino et al. (2016) showed that the simultaneous COS increase during the LIA confirms that the LIA $CO_2$ decline was caused by net terrestrial uptake due to cooling (heterotrophic respiration declining more than Gross Primary Production, due to its higher sensitivity on temperature changes), though a very recent paper estimating the amount of carbon taken up by land use change following the colonisation of the Americas by the Europeans (Koch et al., 2019) provides a different view. Nonetheless, the multi-species approach used by Rubino et al. (2016, e.g. using trends of $CO_2$, $\delta^{13}C$-$CO_2$ and COS) can provide multiple constraints to help understand the biogeochemical processes behind atmospheric $CO_2$ variations over the recent past.*", and a sentence at page 17 lines 23-26: "*The emissions from anthropogenic land use change have also been quantified for each world region (Pongratz and Caldeira, 2012, see Fig. 6e), and can be used to subtract the human contribution from the total $CO_2$ change, even though there is a debate on the amount of land use change following the European colonisation of the Americas (Koch et al., 2019).*"

---

## Author Comment (AC2) · 8 Mar 2019

We thank Referee #1 to support publication of our paper. In the following, we address the referee's comments in the order of appearance in the annotated manuscript.

**Page 2, line 1 (of the manuscript with no track changes revisions)**

Comment from referee: The referee has highlighted the text "(GHGs, including halocarbons)" of the following sentence in the manuscript "The three well-mixed atmospheric greenhouse gases (GHGs, including halocarbons) that contribute the most to current global warming are $CO_2$, $CH_4$ and $N_2O$." and has commented: "**there is a contradiction here as the sentence refers to only three GHGs**."

Author's response: The sentence refers to only three GHGs, as the referee correctly points out, because our study does not include another group of well-mixed GHGs that is halocarbons. We agree with the referee that the sentence could be misleading. We believe there is no need to mention these "synthetic greenhouse gases", as the radiative forcing of even the largest one, CFC-12, was below that of $N_2O$ for all but 1984-2008. $CO_2$, $CH_4$ and $N_2O$ are the long lived GHGs that contribute most to "current" global warming.

Author's changes in manuscript: We have changed the sentence at page 2 lines 1-2 to: "*The three well-mixed (long-lived) atmospheric greenhouse gases (GHGs) that contribute the most to current global warming are $CO_2$, $CH_4$ and $N_2O$.*"

**Page 5, line 2 (of the manuscript with no track changes revisions)**

Comment from referee: "**This paragraph needs a map which depicts the different ice core sites, the accumulation distribution and the wind direction**"

Author's response: We agree with the referee that a map would help the reader, so we have produced a map of Law Dome, which is now figure 1. The other figures numbers have been changed accordingly.

Author's changes in manuscript: A reference to Figure 1 has been added at page 5 line 8 "*The ice cores used in this study, referred to as DE08, DE08-2, DSS and DSS0506, were drilled at Law Dome, East Antarctica (Fig. 1).*" Also, the figure caption for figure 1 describes the map: "*Figure 1: Map of the Law Dome region, slightly modified from Smith et al. (2000), showing the location of the drilling sites DE08, DE08-2, DSS, DSS0506 and DSSW20K discussed in the text. Dotted lines are accumulation isopleths (kg m$^{-2}$ yr$^{-1}$) and unbroken lines are elevation contours (meters above sea level). The inset shows the location of the region in Antarctica*"

**Page 5, line 8 (of the manuscript with no track changes revisions)**

Comment from referee: The referee has highlighted the text "Re-working of the accumulated snow is minimal as high wind speeds are relatively infrequent and the snow surface is smooth." and has commented "**I question this statement, there is always wind re-working. You could, however, say that the wind re-working is not able to erase annual layers**"

Author's response: We agree with the referee that our sentence can be improved to describe wind conditions at Law Dome more accurately.

Author's changes in manuscript: We have modified the sentence at page 5 lines 11-13 to: "*Re-working of the accumulated snow is insufficient to erase annual layers as high wind speeds are relatively infrequent. The resulting annual layering is thick and regular and preserved for much of the ice thickness (van Ommen et al., 2005)*"

**Page 6, line 5 (of the manuscript with no track changes revisions)**

Comment from referee: The referee has highlighted the text "==ice grown with no visible bubbles in it==" and has commented "**this is too vague, how exactly do you produce this ice?**"

Author's response: A detailed explanation of how BFI is produced has been provided in previously published article, such as Rubino et al. (2013). Here, we report a short explanation and introduce a reference to Rubino et al. (2013).

Author's changes in manuscript: We have added a sentence at page 6 lines 12-17 to explain how BFI is produced: "*BFI is grown in ICELAB by keeping a container filled with deionized water in thermal equilibrium, in order to grow ice as slowly as possible from the bottom to the top of the container. The container features Plexiglass sidewalls that are electrically heated. The water exchanges heat only through the metallic base and freezes from the bottom to the top. If the process is slow enough, the produced ice is free of visible bubbles. The results of the tests performed on ICELAB-BFI, as well as on other, externally grown BFI, have been extensively described by Rubino et al. (2013).*"

**Page 6, line 25 (of the manuscript with no track changes revisions)**

Comment from referee: The referee has highlighted the text "==condensable gases and to extract $CO_2$ (plus $N_2O$) from air. The residual $CO_2$ (and $N_2O$) is injected into the MAT252 ion source via==" and has commented "**you have to describe somewhere how you correct for the $N_2O$ interference in the mass spec (chemical slope calibration)**"

Author's response: A detailed explanation of how the $N_2O$ correction is performed has been provided in previously published article, such as Allison and Francey (2007). Here, we report a short explanation citing Allison and Francey (2007).

Author's changes in manuscript: We have added a sentence at page 7 lines 4-7 to explain how the $N_2O$ correction is performed: "*Nitrous oxide ($N_2O$) has identical molecular masses to $CO_2$ and interferes with the isotopic analyses. To remove this interference, a correction is made to the IRMS output in GASLAB using the relative ionisation efficiency of $N_2O$ and $CO_2$, the isotopic composition of $N_2O$ and the measured $N_2O$ and $CO_2$ concentration, as described in detail by Allison and Francey (2007).*"

**Page 8, lines 17-21 and page 11, lines 12-15 (of the manuscript with no track changes revisions)**

Comment from referee: At page 8, line 17-21, the referee has highlighted the text: "==Francey et al. (1999) estimated statistical and systematic $\delta^{13}$C-$CO_2$ biases between 0.025 and 0.07 ‰, and uncertainties of up to ±0.05 ‰, but found an unexplained discrepancy of up to 0.2 ‰ (Trudinger, 2000, section 3.8) around 1900 AD from the South Pole $\delta^{13}$C-$CO_2$ firn record measured at NOAA-INSTAAR (National Oceanic and Atmospheric Administration-Institute of Arctic and Alpine Research, Boulder, Colorado).==" and has commented "**is it worrysome that the Bauska data agree better with the uncorrected Francey data than with the revised data by Rubino (see below)? Note that the Bauska data for the last glacial termination also agree within error limits with the data by Schmitt et al., 2012. This needs more discussion**". At page 11, line 12-15, the referee has highlighted the text: "==It is important to resolve the difference between the Law Dome and the WAIS d13C-CO2 records to establish a Pre-Industrial baseline and, thus, a Pre-Industrial-to-Industrial d13C-CO2 difference, as well as a Pre-Industrial to Last Glacial Maximum $\delta^{13}$C-$CO_2$ difference. These could be useful values 15 for biogeochemical interpretation (Broecker and McGee, 2013; Krakauer et al., 2006).==" and has added a comment which reminds to the comment at page 8, line 17-21: "**see comment above on the offsets between different d13C records**".

Author's response: We completely agree with the referee that it is worrisome that the Bauska $\delta^{13}$C data do not agree with the Rubino et al. $\delta^{13}$C dataset, which has revised the Francey et al. $\delta^{13}$C dataset and resolved the discrepancy between the Law Dome and South Pole $\delta^{13}$C records. Because of the way we have decided to structure our paper (section 3.1 shows the old Law Dome datasets, whereas section 3.2 shows the new Law Dome records and how they compare with records from other sites), we believe it is more appropriate to add the discussion suggested by the referee in section 3.2 at page 11 lines 24-32.

Author's changes in manuscript: We have modified the discussion at page 11 line 33 – page 12 line 10 to address the referee's comment: "*The Bauska et al. (2015) record agrees within uncertainties with the Francey et al. (1999) dataset. However, Rubino et al. (2013) is the only record to show consistency with all firn records and direct atmospheric measurements (see Fig.s 3c and 5a). This would suggest that the Rubino et al. (2013) is currently the most accurate record and should be used to set a pre-industrial baseline. However, no definite conclusion can be drawn until a thorough intercomparison study is carried out between the labs that have produced the WAIS and the Law Dome $\delta^{13}$C-CO$_2$ datasets (Oregon State University-University of Colorado-Institute of Arctic and Alpine Research, INSTAAR and CSIRO). It is important to resolve the difference between the Law Dome and the WAIS $\delta^{13}$C-CO$_2$ records in order to establish a Pre-Industrial baseline and, thus, a Pre-Industrial-to-Industrial $\delta^{13}$C-CO$_2$ difference. Setting a Pre-Industrial baseline could have consequences on the Last Glacial Maximum-to-Pre-Industrial $\delta^{13}$C-CO$_2$ difference as well (Schmitt et al., 2012). These values are useful for biogeochemical interpretation (Broecker and McGee, 2013; Krakauer et al., 2006).*"

**Page 9, line 1-2 (of the manuscript with no track changes revisions)**

Comment from referee: The referee has highlighted the text: "In doing so, they resolved the 0.2 ‰ discrepancy found between the Law Dome $\delta^{13}$C-CO$_2$ record and the South Pole $\delta^{13}$C-CO$_2$ firn record." and has commented: "**these d13C offsets are the most important discrepancies, it may be worthwhile to show the South Pole data**"

Author's response: We agree with the referee that the $\delta^{13}$C difference between the Law Dome ice core record and the South Pole firn record was the most important discrepancy found between datasets from different sites. However, we feel that this issue has been extensively discussed in the Rubino et al. (2013) paper. Also, showing the South Pole firn data in figure 2 (where the only other firn record is shown, now figure 3) would be out of context, as the figure only shows ice and firn data from Law Dome. Therefore, we have added an extra plot (figure 5) with the South Pole firn data and a reference to the Rubino et al. (2013) paper for further details.

Author's changes in manuscript: We have added a line at page 9, line 16-18: "*In doing so, they resolved the 0.2 ‰ discrepancy found between the Law Dome $\delta^{13}$C-CO$_2$ record and the South Pole $\delta^{13}$C-CO$_2$ firn record (the South Pole firn records have been reported in Fig 5, but see Rubino et al., 2013, for more details)*"

**Page 9, lines 14-23 (of the manuscript with no track changes revisions)**

Comment from referee: The referee has highlighted the text:

"- To investigate changes in Pre-Industrial sources of CH$_4$, Ferretti et al. (2005) produced a record of $\delta^{13}$C-CH$_4$ in Law Dome ice covering the last 2000 years (not shown). They reported unexpected changes of the global CH$_4$ budget, mainly attributed to variations of biomass burning emissions during the Late Pre-Industrial Holocene (LPIH) through an atmospheric box model (Lassey et al.,

2000). The $\delta^{13}$C-CH$_4$ record from Ferretti et al. (2005) has not been included in ICEBASE because the air samples extracted in ICELAB were measured on a mass spectrometer not maintained by CSIRO-GASLAB. Therefore, the $\delta^{13}$C-CH$_4$ data are not on a CSIRO calibration scale.

- Park et al. (2012) measured oxygen and intramolecular nitrogen isotopic compositions of N$_2$O (not shown) covering 1940 to 2005 in Law Dome firn air and archived air samples from Cape Grim (Tasmania). In doing so, they confirmed that the rise in atmospheric N$_2$O levels is largely the result of an increased reliance on nitrogen-based fertilizers. These measurements are not included in ICEBASE either.” and has commented “**as this data base summarizes all gas results from Law Dome, it is really a pity that the Mischler and Park data are not accessible via this tool as well. It is clear that these data cannot be updated constantly, as they come from other labs, but the status quo of the data sets as published could be included in the data collection**”

Author's response: It is tempting to include the $\delta^{13}$C-CH$_4$ and $\delta^{15}$N-N$_2$O records in our database, but, as the referee has correctly points out, those data have not been produced in CSIRO GASLAB. Therefore, we have no control over them. We have written this paper to make the scientific community aware of the updates we have performed on our CO$_2$/$\delta^{13}$C-CO$_2$, CH$_4$ and N$_2$O dataset, and to explain how the dataset has been produced and revised. One of the main reasons to revise and publish these records was because our CO$_2$, CH$_4$ and N$_2$O data were used in the Meinshausen et al. (2017) compilation and $\delta^{13}$CO$_2$ in the Graven et al compilation for CMIP6 (we say this in the Introduction), whereas we are not aware of CH$_4$ or N$_2$O isotopes being used in CMIP6. The risk of introducing records that are not treated in the same way as the CSIRO data is that we raise confusion among people using the datasets. The Ferretti (not Mischler as the referee writes) and Park data can be downloaded from different repositories and will always be available to the scientific community, independently from the updates of the Law Dome datasets we will release from time to time.

**Page 10, line 11 (of the manuscript with no track changes revisions)**

Comment from referee: The referee has highlighted the text: “Comparison with records from other cores show general agreement, but also a number of unexplained discrepancies.” and has commented “**too vague**”

Author's response: This sentence is vague because it is just an introduction to the following list of bullet points. We has replaced it with a clearer sentence.

Author's changes in manuscript: We have modified the sentence at page 10 line 27: “*The following list compares the new Law Dome records with records from other sites and discusses the main differences:*”

**Page 10, lines 30-32 (of the manuscript with no track changes revisions)**

Comment from referee: The referee has highlighted the text: “The difference is even more surprising when the tight agreement between the Law Dome CH$_4$ record and the WAIS CH$_4$ record (Mitchell et al., 2011) around this time is considered (compare red circles and grey squares in Fig. 3c).” and has commented “**I don't really understand this argument. The CO$_2$ and CH$_4$ measurements are independent and in case of Mitchell even the gas extraction is separate from that of CO2**”

Author's response: The CO$_2$ and CH$_4$ measurements are mostly independent from each other in terms of measurement technique, but their comparison allows us to test whether the two sites of Law Dome and WAIS are providing consistent records in terms of smoothing of the atmospheric signal. The fact that the CH$_4$ records are consistent, whereas the CO$_2$ records show significant

differences points towards a problem with $CO_2$, rather than issues related to poor understanding of the site (dating, firn smoothing, etc...). In other words, the consistency between the Law Dome and the WAIS $CH_4$ records is evidence of

1. Consistent dating (ice age and gas age)
2. Similar smoothing of the atmospheric signals at the two sites ([ice age-gas age and gas age distribution).

Therefore, the differences found in the $CO_2$ records are hard to explain with issues of dating uncertainty, as partly claimed by Ahn et al. (2012), or differences in smoothing of the atmospheric signals between the two sites. This support a chemical origin of the discrepancies (in-situ production), which is more likely to occur for $CO_2$ than for $CH_4$.

That said, the comment of the reviewer has alerted us that the way we have phrased our explanation is not very clear. Therefore, we have modified the text to make it easier to understand.

Author's changes in manuscript: We have modified the sentence a line at page 11 lines 16-18, by adding: "*The consistency between the Law Dome and the WAIS $CH_4$ record rules out dating issues or large differences in smoothing of the atmospheric signals between the two sites. This suggests a chemical origin (in-situ production) of the $CO_2$ discrepancies, which is more likely to occur for $CO_2$ than for $CH_4$.*"

**Page 11, lines 18-21 (of the manuscript with no track changes revisions)**

Comment from referee: The referee has highlighted the text: "==Interestingly, the LIA $CH_4$ decrease measured at NEEM appears to start before the $CH_4$ decrease measured at Law Dome/WAIS, suggesting that the LIA event had an effect on the Northern Hemisphere $CH_4$ concentration first, and then propagated to the Southern Hemisphere==." and has commented "**Is such a multidecadal lag really possible from a $CH_4$ cycle/atmospheric mixing point of view? You could check this using a multi-box model. It looks more like an age scale issue as the Mitchell record is shifted to younger ages compared to Law Dome in this time interval.**"

Author's response: We thank the referee for this important comment, which has compelled us to think about this issue carefully. We have updated the original NEEM $CH_4$ gas age scale published in Rhodes et al. (2013), with the new NEEM ice age scale published in Sigl et al. (2015) and the revised delta_age (gas age-ice age) after Buizert et al. (2014). The different timing is now partially resolved (15 years now, instead of 30 years). Having added the $CH_4$ GISP2 record (as suggested by reviewer 2), we have commented on how the Northern (GISP2) and Southern (WAIS) records have been synchronised by Mitchell et al. (2013) based on the reasoning that "The multidecadal events observed in both ice core records must have occurred simultaneously since the durations of the events were much larger than the atmospheric mixing time (~1 year)." There are multiple possible reasons, associated with the differences of the sites and/or the sampling resolution of the records, to explain the discrepancy found between Northern (NEEM) vs Southern (Law Dome/WAIS) Hemisphere $CH_4$ records, as discussed in the following:

- Age scale issues, as suggested by the referee. Usually, age scale issues are not very significant for the last centuries. Over such a short time scale, the age of the ice is established through annual layer counting. However, there can be significant uncertainty associated with the $\Delta$age (ice age-gas age).
- Smoothing of the atmospheric signals due to air diffusion in the firn open porosity. As shown for the Law Dome $CO_2$ record smoothed through the gas age distribution of DML in Figure 2

of Rubino et al. (2016), the smoothing causes a shift in age of the max and min values of the LIA $CO_2$ decrease. For the Law Dome vs DML comparison, the shift amounts to a few years. NEEM has higher accumulation than DML, so firn smoothing should cause an even smaller shift.

- Inadequate sampling resolution. The NEEM $CH_4$ records plotted in figure 3 (now figure 4) is a 5 year average of the high resolution record published in Rhodes et al. (2013). The Law Dome $CH_4$ record has a lower sampling resolution (5 datapoint over a 30-year period of decreasing $CH_4$ concentration, so on average 1 datapoint every 6 years). By increasing the sampling resolution, it is possible that the time shift between the NEEM and the Law Dome $CH_4$ event decreases.

A thorough investigation of the cause of the differences in the LIA $CH_4$ event in the Northern (as recorded in NEEM) and the Southern (as recorded in Law Dome/WAIS) Hemisphere is out of the scope of our paper. However, in the future, this discrepancy should be resolved to obtain a precise synchronisation of all ice core records available over the LIA.

Author's changes in manuscript: We have modified the discussion at page 12 lines 18-27 (now page 12, lines 10-20): "*The age scale of the NEEM $CH_4$ record published in Rhodes et al. (2013) has been revised with the updated ice age scale published in Sigl et al. (2015) and the new estimate of Δage provided by Buizert et al. (Buizert et al., 2014). Mitchell et al. (2013) have synchronised the GISP2 $CH_4$ record with the WAIS $CH_4$ record to investigate changes of the Inter Polar Difference in the Pre-Industrial based on the reasoning that "the multidecadal events observed in both ice core records must have occurred simultaneously since the durations of the events were much larger than the atmospheric mixing time (~1 year)" (Mitchell et al., 2013). The NEEM $CH_4$ record has not been synchronised with the others, and there are multiple possible reasons, including age scale issues, different smoothing of the atmospheric signals at the different sites and inadequate sampling resolution, to explain the discrepancy found between the NEEM and the GISP2/Law Dome/WAIS $CH_4$ records during the LIA. A thorough investigation is out of the scope of this paper, but, in the future, this discrepancy should be resolved to obtain a precise synchronisation of all ice core records available over the LIA.*".

**Page 12, lines 20-25 (of the manuscript with no track changes revisions)**

Comment from referee: The referee has highlighted the text: "Considering that:

- All GHG records in ice cores are a smoothed representation of the real atmospheric history;

- DSS is the highest accumulation rate site ever sampled in Antarctica recording the LIA $CO_2$ event;

- There is the risk that the WAIS core is affected by in-situ production of $CO_2$;

- Accurate CO2 record have not been derived from Greenland ice cores

we suggest that there is a need to sample a new, clean and deep ice core from Law Dome, to recover the real atmospheric LIA $CO_2$ decrease and other rapid changes in atmospheric composition during Pre-Industrial millennia." and has commented "**this would fit better into the conclusions**"

Author's response: We agree with the referee.

Author's changes in manuscript: We have moved the sentence from page 12 lines 20-25 to page 18 lines 16-23.

**Page 13, lines 2-3 (of the manuscript with no track changes revisions)**

Comment from referee: The referee has highlighted the text: "==processes removing GHGs from the atmosphere (sinks)==." and has commented "**typically we do not talk about sinks in the case of CO₂ as for the carbon cycle there is not really a destruction of the molecule but a constant exchange of carbon between different reservoirs**"

Author's response: We agree with the referee that since there is no destruction of $CO_2$ in the atmosphere, then, strictly speaking, there is no "sink" in the case of $CO_2$. However, the general biogeochemical literature uses the word "sink" to signify a process removing GHGs, including $CO_2$, from the atmosphere (e.g. information from the Global Carbon Project https://www.globalcarbonproject.org/). Thus, we have decided to leave the sentence unaltered.

**Page 13, line 29 (of the manuscript with no track changes revisions)**

Comment from referee: The referee has highlighted the text: "==recent==" and has commented "**unclear**"

Author's response: To clarify, the word "recent" here means over the last decades, as opposed to the records from ice cores covering the last centuries/millennia.

Author's changes in manuscript: We have replace the word "recent" now at page 15 line 4 with "*over the last decades*".

**Page 14, lines 21-22 (of the manuscript with no track changes revisions)**

Comment from referee: The referee has highlighted the text: "==tropospheric species [OH], oxidation by stratospheric species [OH, Cl and O(1D)], and oxidation in soils are the main sinks==." and has commented "**marine boundary layer Cl sink**"

Author's response: We thank the reviewer to have reminded us of this sink.

Author's changes in manuscript: We have added the text "*and reactive chlorine in the marine boundary layer*" at page 15 line 33.

**Page 14, line 33 (of the manuscript with no track changes revisions)**

Comment from referee: The referee has highlighted the text: "==hypothesis of an early anthropogenic influence==." and has commented "**the use of this expression is misleading here, as the "early anthropogenic influence hypothesis" by Ruddiman refers to a much earlier increase. Please change the wording.**"

Author's response: Generally, the "early anthropogenic hypothesis" refers to the explanation of an influence on the changes of atmospheric $CO_2$ and $CH_4$ concentration during the Holocene, originally suggested by Ruddiman. In some of his papers, Ruddiman also discusses the influence of human activities on the atmospheric $CO_2$ and $CH_4$ concentration during the Little Ice Age. Therefore, the "early anthropogenic hypothesis" can also refer to any pre-industrial alteration of the atmospheric chemical composition caused by human activities. However, to avoid confusion, we follow the referee's suggestion.

Author's changes in manuscript: We have modified the text at page 16 line 11 to "*supporting the hypothesis of a pre-industrial anthropogenic influence on atmospheric $CH_4$*".

**Page 15, lines 9-10 (of the manuscript with no track changes revisions)**

Comment from referee: The referee has highlighted the text: "As already mentioned, the Pre-Industrial inter-hemispheric $N_2O$ difference is also poorly constrained." and has commented "**I doubt this can be resolved, as it is extremely small due to the long life-time of $N_2O$**"

Author's response: We agree with the referee that the Inter-hemispheric $N_2O$ difference is small and we would need very high precision ice core measurements to resolve it. However, with the development of new analytical techniques, it could become feasible. Therefore, we have decided to leave the sentence unaltered

**Page 16, lines 1-4 (of the manuscript with no track changes revisions)**

Comment from referee: The referee has highlighted the text: "There are regional (Mann et al., 2008), continental (Pages2k, 2013) and hemispheric (Neukom et al., 2014) temperature reconstructions that can be used in Coupled Carbon Cycle-Climate Models to quantify the contribution from each region to the total $CO_2$ decrease (Fig. 4b)." and has commented "**the wording is weird. It is not clear how the reconstructions could be "Used in Coupled Climate Models", the provide a benchmark**"

Author's response: We are just suggesting that regional temperature reconstructions could be used as "forcing" for models describing the relationship between climate and carbon cycle to quantify the regional contribution to the total $CO_2$ change recorded through the LIA

Author's changes in manuscript: We have modified the sentence at page 17 line 17: "*There are regional (Mann et al., 2008), continental (Pages2k, 2013) and hemispheric (Neukom et al., 2014) temperature reconstructions that can be used to drive models describing the relationship between climate and carbon cycle to quantify the contribution from each region to the total $CO_2$ decrease*".

**Page 16, lines 11-13 (of the manuscript with no track changes revisions)**

Comment from referee: The referee has highlighted the text: "The consequences of the LIA climatic changes on contemporary societal development are important for understanding why different communities were more or less vulnerable, resilient or even adaptive (Degroot, 2018), and plan future choices accordingly." and has commented "**This is a little far fetched as all societies today have technical means that were not available during the LIA. So it can be questioned, whether we really learn something from the LIA about societal resilience.**"

Author's response: We agree with the referee that current societies have more advanced technical means than in the LIA. However, the ability of a society to adapt to climate change will not depend on technical means only. Political and socio-economic factors will influence how efficiently our society will adapt to future climate change. So, being able to quantify which regions have been more vulnerable to past climate change, also in terms of the response of the natural carbon cycle, could help plan future adaptation strategies.

Author's changes in manuscript: To put our statement into perspective, we have modified the sentence at page 17 lines 29-31: "*The consequences of the LIA climatic changes on societal development are important for understanding why different communities were more or less vulnerable, resilient or even adaptive (Degroot, 2018). Being able to quantify which regions have been more vulnerable to past climate change, also in terms of the response of the natural carbon cycle, could help plan future adaptation strategies*".

**Page 16, lines 15-16 (of the manuscript with no track changes revisions)**

Comment from referee: The referee has highlighted the text: "($\delta^{13}$C-CO$_2$, $\delta^{13}$C-CH$_4$, $\delta$D-CH$_4$, $\delta^{15}$N-N$_2$O, $\delta^{18}$O-N2O)" and has commented "**here you refer to the isotopes of all greenhouse gases, but you only store $\delta^{13}$CO$_2$ in your data base (see comment above).**"

Author's response: A few lines below (page 16, lines 17-18) we have clearly stated that, of all the isotopic records, only the $\delta^{13}$C-CO$_2$ is constantly being updated and revised. However, to avoid confusion, we have removed the records that are not stored in the database.

Author's changes in manuscript: We have modified the sentence at page 18, lines 2-3: "*The records of GHG (CO$_2$, CH$_4$, N$_2$O) concentrations and the isotopic composition of CO$_2$ ($\delta^{13}$C-CO$_2$) from the Law Dome ice cores are one of the most important sources of information for models trying to predict the future behaviour of biogeochemical cycles and their influence on the climate system.*"

**Supplement page 2**

Comment from referee: The referee has highlighted the text: "derive the gas-age from the ice-age-vs-gas-age difference for ice samples and gas-age vs depth for firn samples" and has commented "**it is not clear how this is done (firn modeling). Please elaborate on this somewhere in the supplement**"

Author's response: An extensive description of how gas age is attributed to ice and firn samples has been provided in past papers. For firn, the dating is based on the firn model, but for ice samples the gas-age – ice age is based on a number of factors, including firn modelling.

Author's changes in the Supplement: We have added the following line to the text at page 2: "*For firn samples, the dating is based on the firn air diffusion model, whereas for ice samples it is based on a number of factors, including firn model (Trudinger et al., 2013)*".

**Supplement page 4**

Comment from referee: The referee has highlighted the text: "The blank correction is quantified by the average deviation of replicated BFI/Blanks measured concentration and isotopic composition from the expected value (i.e.: the value associated with the reference gas used). A blank correction is calculated for each period when the conditions of preparation/storage/extraction are the same. In other words, a new blank correction is calculated each time any of the factors (operator, freezer, temperature of cold room, duration of extraction, etc...) that are believed to influence preparation/storage/extraction changes. The blank correction has an uncertainty associated with it, given by the standard deviations of differences from the expected value." and has commented: "**how large is this blank correction typically for the different measured species**"

Author's response: We agree with the referee that it is useful for the reader to have an idea of the size of the blank correction, so we have added the best estimate for each species.

Author's changes in the Supplement: We have added a sentence at page 4: "*The blank correction can vary significantly depending on the conditions of the extraction line and on how experienced the line operator is. Typical values are within the following ranges: 0.5-1.5 ppm (uncertainty 0.5-2 ppm, 1$\sigma$) for CO$_2$, 3-10 ppb (uncertainty 3-15 ppb, 1$\sigma$) for CH$_4$, 0.5-3 ppb (uncertainty 0.5-4 ppb, 1$\sigma$) for N$_2$O and 0.03-0.1 ‰ (uncertainty 0.04-0.13 ‰, 1$\sigma$) for $\delta^{13}$C-CO$_2$.*"

**Supplement page 4**

Comment from referee: The referee has highlighted the text: "The gravity correction: Gravitational enrichment of heavier species in air in the firn open porosity (Craig et al., 1988; Schwander et al.,

1988) has different effects depending on the difference between the mass of the measured species and the average mass of air: [X]corr = 10-3 × d15N × (MX - Mair) × [X]meas, where X is the measured species (i.e.: $CH_4$, $CO_2$ and $N_2O$), "corr" and "meas" stands for corrected and measured respectively, and $\delta^{15}N$ is the isotopic ratio of molecular nitrogen ($N_2$) in firn. For $\delta^{13}C$ measurements, the gravity corrected $\delta^{13}C$ equals the sum of a correction factor (very close to the $\delta^{15}N$-$N_2$) and the measured $\delta^{13}C$ (see Rubino et al., 2013 for details)." and has commented: "**how large is this gravity correction typically for the different species? Here you refer to $\delta^{15}N_2$ values. Where do they come from? Are these values also stored in the database (they should)? How do you correct iff you do not have $\delta^{15}N_2$ values?**"

Author's response: The questions of the referee are all very relevant and we have included a few lines to answer them in the Supplement.

Author's changes in the Supplement: We have added a few lines at page 4: "*The values of $\delta^{15}N$ are often measured in firn to constrain the firn diffusion model. In case of missing $\delta^{15}N$ measurements, the firn model is constrained with other measurements and then used to simulate the $\delta^{15}N$ profile. Our database stores all measured and modelled $\delta^{15}N$ values for firn sites at Law Dome. The gravity correction is typically 1-1.5 ppm for $CO_2$, 2-6 ppb for $CH_4$, 1-1.4 ppb for $N_2O$ and 0.25-0.3 ‰ for $\delta^{13}C$-$CO_2$.*"

**Supplement page 4**

Comment from referee: The referee has highlighted the text: "The diffusion correction (only for measurements of isotopic composition): For measurements of isotopic ratios in firn and ice air samples, a so-called diffusion correction is needed (Trudinger et al., 1997). This correction arises from the fact that an isotope ratio is the ratio of two isotopes with slightly different diffusion coefficients and therefore slightly different effective ages (Trudinger, 2000, section 3.6). For hypothetical species with constant isotopic ratio, but changing atmospheric concentrations, the isotopic ratio in the firn can be significantly different from the atmospheric ratio. For $\delta^{13}C$, the diffusion correction is proportional to the rate of change of $CO_2$ concentration, which makes the $\delta^{13}C$ diffusion correction insignificant in the LHPI, and a very significant term in the Industrial Period." and has commented "**how large is this correction typically? How do you do the correction (firn modeling)?this needs more detail**"

Author's response: The questions of the referee are all very relevant and we have included a few lines to answer them in the Supplement.

Author's changes in the Supplement: We have added a few lines at page 4: "*The diffusion correction is estimated using the CSIRO firn diffusion model and, at Law Dome, can range from around 0 in the Pre-Industrial to 0.13 ‰ in the Industrial Period.*"

---

## Author Comment (AC3) · 8 Mar 2019

We thank Referee #2 to support publication of our paper. In the following, we address the referee's comments in the order of appearance.

**Page 2, line 15-16 (of the manuscript with no track changes revisions)**

Comment from referee: **I suggest to rephrase "it is extremely difficult to separate the impacts of anthropogenic increases in CO₂ on carbon sinks from the impacts of global warming or increased CO₂ concentration on these sinks.". This statement was unclear for me**

Author's response: We have change the statement to make it clear.

Author's changes in manuscript: We have modified the sentence at page 2 lines 15-17 to: "*Additionally, temperature and $CO_2$ have both increased almost continuously through the 20th century, making it difficult to separate the impacts of $CO_2$ on carbon sinks from the impacts of temperature increase on these sinks.*"

**Page 3, line 7 (of the manuscript with no track changes revisions)**

Comment from referee: **Rhodes et al., 2016 (Climate of the Past) include more CH4 data from Greenland.**

Author's response: We thank the referee for this important comment, which has helped us address an important weakness of our study when comparing the Law Dome $CH_4$ records with $CH_4$ records from other sites. As the referee points out, Rhodes et al. (2013) shows the GISP2 $CH_4$ record as well, published in Mitchell et al. (2013). We have added this record to figure 3c (now 4c) and have cited Mitchell et al. (2013).

Author's changes in manuscript: We have modified the sentence at page 3 lines 8 adding "… and from GISP2 (Mitchell et al., 2013)…" and have added the GISP2 $CH_4$ record to figure 3c (now 4c).

**Page 5, section 2.1**

Comment from referee: **use the same unit for all accumulation data, for homogeneity**.

Author's response: We agree with the reviewer that it is better to have all accumulation rates expressed with the same units. We have decided to convert all accumulation rates to kg m$^{-2}$ yr$^{-1}$, which is also consistent with the units used in Figure 1. We have also left the accumulation rate at DE08/DE08-2 as meters of ice equivalent per year in brackets, as it is nice to give a better feel to the non-specialist reader for what the accumulation looks like.

Author's changes in manuscript: We have expressed all units of accumulation rate at page 5 in kg m$^{-2}$ yr$^{-1}$.

**Page 6, line 23 (of the manuscript with no track changes revisions)**

Comment from referee: typo "I think".

Author's response: fixed

**Page 11, line 20 (of the manuscript with no track changes revisions)**

Comment from referee: **I understand that the atmospheric mixing between Northern hemisphere and Southern hemisphere is fast enough so [CH4] would exhibit almost simultaneous trend in both hemispheres. Here the shift in in LIA CH4 decrease seems to be about 40 yrs (Fig. 3). Similar shift**

**seems to exist at the onset of the industrial period CH4 increase. Can we explain such shift with Age Scale uncertainty? Maybe discuss this shift by providing more quantitative estimation**.

Author's response: We thank the reviewer for this important comment, which, together with a similar comment of reviewer 1, and has helped us improve this part of the manuscript. We have updated the original NEEM CH$_4$ gas age scale published in Rhodes et al. (2013), with the new NEEM ice age scale published in Sigl et al. (2015) and the revised Δage (gas age-ice age) after Buizert et al. (2014). The different timing is now partially resolved (15 years now, instead of 30 years). Having added the CH$_4$ GISP2 record, we have commented on how the Northern (GISP2) and Southern (WAIS) records have been synchronised by Mitchell et al. (2013) based on the following reasoning: "The multidecadal events observed in both ice core records must have occurred simultaneously since the durations of the events were much larger than the atmospheric mixing time (~1 year).". As already written above, there are multiple possible reasons, associated with the differences of the sites and/or the sampling resolution of the records, to explain the discrepancy found between Northern (NEEM) vs Southern (Law Dome/WAIS) Hemisphere CH4 records, as discussed in the following:

- Age scale issues, as suggested by the referee. Usually, age scale issues are not very significant for the last centuries. Over such a short time scale, the age of the ice is established through annual layer counting. However, there can be significant uncertainty associated with the delta_age (ice age-gas age).
- Smoothing of the atmospheric signals due to air diffusion in the firn open porosity. As shown for the Law Dome CO2 record smoothed through the gas age distribution of DML in Figure 2 of Rubino et al. (2016), the smoothing causes a shift in age of the max and min values of the LIA CO2 decrease. For the Law Dome vs DML comparison, the shift amounts to a few years. NEEM has higher accumulation than DML, so firn smoothing should cause an even smaller shift.
- Inadequate sampling resolution. The NEEM CH4 records plotted in figure 3 (now figure 4) is a 5 year average of the high resolution record published in Rhodes et al. (2013). The Law Dome CH4 record has a lower sampling resolution (5 datapoint over a 30-year period of decreasing CH4 concentration, so on average 1 datapoint every 6 years). By increasing the sampling resolution, it is possible that the time shift between the NEEM and the Law Dome CH4 event decreases.

A thorough investigation of the cause of the differences in the LIA CH4 event in the Northern (as recorded in NEEM) and the Southern (as recorded in Law Dome/WAIS) Hemisphere is out of the scope of our paper. However, in the future, this discrepancy should be resolved to obtain a precise synchronisation of all ice core records available over the LIA.

Author's changes in manuscript: We have modified the discussion at page 12 line 18-27: "*The age scale of the NEEM CH$_4$ record published in Rhodes et al. (2013) has been revised with the updated ice age scale published in Sigl et al. (2015) and the new estimate of Δage provided by Buizert et al. (Buizert et al., 2014). Mitchell et al. (2013) have synchronised the GISP2 CH$_4$ record with the WAIS CH$_4$ record to investigate changes of the Inter Polar Difference in the Pre-Industrial based on the reasoning that "the multidecadal events observed in both ice core records must have occurred simultaneously since the durations of the events were much larger than the atmospheric mixing time (~1 year)" (Mitchell et al., 2013). The NEEM CH$_4$ record has not been synchronised with the others, and there are multiple possible reasons, including age scale issues, different smoothing of the atmospheric signals at the different sites and inadequate sampling resolution, to explain the*

*discrepancy found between the NEEM and the GISP2/Law Dome/WAIS CH$_4$ records during the LIA. A thorough investigation is out of the scope of this paper, but, in the future, this discrepancy should be resolved to obtain a precise synchronisation of all ice core records available over the LIA.".*

**Figures**:

Comment from referee: **I find the figures difficult to read: I would advise to increase size for labels and titles.**

Author's response: We have chosen the size for labels and titles based on the font size in the template provided for ESDD articles for consistency. We would be happy to increase the size if the editorial office allows us to do so.

Author's changes in manuscript: We have not changed the size because we need approval from the ESDD editorial office.

**Figure 4**:

Comment from referee: **this figure highlight two important past findings (Rubino et al., 2016 ; Ferreti et al., 2005). It also includes data that can potentially be relevant for further studies and interpretations. I am not sure these data (b, d, and e) need to be plotted; likely description in the manuscript is enough. If the authors want to keep these data as part of plot 4, I would recommend to clarify the figure, e.g. the panel b and c can be shifted, so the figure reports first ice core data, and second complementary climatic data**.

Author's response: We agree with the reviewer that swapping figures 4b and 4 c (now 5b and 5c) makes sense for clarity of presentation, but also under the point of view of citation order.

Author's changes in manuscript: We have modified figure 4 (now figure 5) by swapping figure 4b and 4c (now 5b and 5c). We have modified the figure caption and the citation to figure 5b and 5c in the text accordingly.

**Technical comments related to Supplement:**

Comment from referee: **My main concern is about the way uncertainties are calculated by multiplying the blank uncertainty with a factor u.f. I find this process complex, not fully understandable for data users, and maybe operator-dependent. To me, the blank uncertainty is independent from other sources of uncertainties (e.g., dispersion of results observed for replicated measurements on of the same sample). Independent uncertainties when propagated do not multiply each others**.

Author's response: We thank the referee for the detailed comments provided on the way we calculate the uncertainty associated with the results we produce. We agree with the referee that the process we have developed is quite complex. Unfortunately, our rule-based selection procedure needs to take into account all the little pieces of information that are normally evaluated by an operator based on the experience matured over 20+ years at ICELAB-GASLAB. Our aim, though, was to develop an automatic system, which would minimise operator-dependent judgements, to make it consistent in time and between different operators. We understand that it is very difficult for an external data user to understand the details of the procedure we have developed. We have tried to make the description of it as clear as possible.

As the referee states, the blank uncertainty is independent from other sources of uncertainties, such as those mentioned by the referee. We have propagated the uncertainty using the guidelines from

the Joint Committee for Guides in Metrology (2008). The uncertainty associated with the blank correction dominates over the other sources of uncertainty.

Author's changes in the Supplement: We have added a citation to the Joint Committee for Guides in Metrology (2008), Evaluation of measurement data—Guide to the expression of uncertainty in measurement, Bur.Int. des Poids et Mesures Pavillon de Breteuil, France, at page 5. We have also added a couple of sentences at page 4: "*The uncertainty associated with gravity and the diffusion corrections are negligible compared to the uncertainty associated with the blank correction*" and at page 5 "*However, the variability associated with replicates of the same sample is negligible compared with the uncertainty associated with the blank correction.*" to explain that the blank uncertainty dominates over the other sources of uncertainty.

Comment from referee: **We can observe here that for a sample where u.f. = 1 (i.e., qf and mq = fair or good), the data uncertainty is reduced to only the blank uncertainty, ignoring for example that different replicated measurements of the same sample will likely not be exactly all the same**.

Author's response: The referee is right: different replicated measurements of the same sample will not show the same result. However, the variability associated with replicates of the same sample is negligible compared to the uncertainty associated with the blank correction, which quantifies the variability associated with multiple tests run together with the samples. Additionally, because of the way the ICELAB-GASLAB system has been conceived (measurements of different species on the same ice sample), replicates of the same sample measured for the same species are so rare that there is no uncertainty associated with replicates for the vast majority of the samples analysed.

Author's changes in the Supplement: We have modified a sentence at page 3: "*Only a few measurements of concentration have been replicated. For isotopes, there has never been enough air to measure replicates*". We have also modified a sentence at page 4 of the Supplement: "*Because of the small size of ice core samples generally available and the need for large air volume to measure multiple species, the vast majority of samples have no replicated measurements.*" and another sentence at page 5: "*However, the variability associated with replicates of the same sample is negligible compared with the uncertainty associated with the blank correction.*"

Comment from referee: **The u.f. factor includes many parameters (flags, and criteria associated to weights), and some of them are not related to uncertainty. As an example, the first parameter is "melt layer". A melt layer can, e.g., results in high methane concentration (due to in situ production), but the measurement uncertainty of such high concentration should not be different from a regular sample. Just higher concentration will be measured. A melt layer sample does not have the quality required for reconstructing past CH$_4$, but its measurement could be of great quality! Overall, I would advise that the authors identify more clearly what causes uncertainties, instead of considering everything.**

Author's response: Our database is in constant development and it is possible that we will find better ways of including factors affecting the total uncertainty in the future. However, for now, when a melt layer is found, a sample is rejected because, as the referee suggests, melt layers are known to alter the concentration of the species we measure.

Author's changes in the Supplement: We have added a sentence at page 3: "*For example, a melt layer is classified as evidence of a fatal problem of sample quality and provides a q.s. = 3 and a q.f. = "reject".*"

Comment from referee: **Some of the flags and criteria associated to weights seem subjective, and maybe operator-dependent in their evaluation (at least this is what I feel when reading the Supplement).**

Author's response: The thresholds for flags and weights have been decided based on a "calibration". The idea was to replicate with an automatic procedure what had been done in the past manually. The aim was to make the procedure consistent over time and as independent from the operator as possible.

Author's changes in Supplement: We have added a line at page 3: "*The flagging and weighting thresholds are tuned by calibrating the rule-based selection on the manual selection used before the database was conceived. The idea was to replicate with an automatic procedure what had been done in the past manually. The aim was to make the procedure consistent over time and as independent from the operator as possible. In summary, the rule-based selection converts qualitative judgments on the robustness of sample preparation, extraction and analysis into quantitative scores in order to consistently select/reject the results and quantify uncertainty.*"

Comment from referee: **When all criteria reports "reject" (i.e., u.f. = 4), the data is not rejected, but the uncertainty is increased more. It seems to me that these data should be excluded (as suggested by the wording "fatal problem").**

Author's response: When q.f. = "reject", the sample is marked with a rejection flag. However, the result of that sample is retained in the database and a u.f. = 4 is associated with it. The reason why we have decided to do that is to give enough flexibility for the operator to overrule the automatic selection, providing a convincing justification that the operator's choice is appropriate. If a data user wants to use that sample, the uncertainty associated with the results is very high (4 × the blank uncertainty and then propagated as explained).

Author's changes in the Supplement: We have added a line at page 5: "*When q.f. = "reject", the sample is marked with a rejection flag. However, the result of that sample is retained in the database and a u.f. = 4 is associated with it. The reason why we have decided to do that is to provide enough flexibility for the operator to overrule the automatic selection when there is a strong reason (e.g. new insight developed over time). If a data user wants to use that sample, the uncertainty associated with the results is very high (4 × the blank uncertainty and then propagated as explained below).*"

Comment from referee: **What are the typical blanks observed, and typical blank uncertainties observed?**

Author's response: This comment, together with a similar comment from referee 1, has helped us provide important information, which was missing from the previous version of the manuscript. We thank the referees for that.

Author's changes in the Supplement: We have added a line at page 5: "*The blank correction can vary significantly depending on the conditions of the extraction line and on how experienced the line operator is. Typical values are within the following ranges: 0.5-1.5 ppm (uncertainty 0.5-2 ppm, $1\sigma$) for $CO_2$, 3-10 ppb (uncertainty 3-15 ppb, $1\sigma$) for $CH_4$, 0.5-3 ppb (uncertainty 0.5-4 ppb, $1\sigma$) for $N_2O$ and 0.03-0.1 ‰ (uncertainty 0.04-0.13 ‰, $1\sigma$) for $\delta^{13}C\text{-}CO_2$.*" and a sentence at the end of page 4 of the Supplement: "*The uncertainty associated with gravity and the diffusion corrections are negligible compared to the uncertainty associated with the blank correction*"

Comment from referee: **I am not convinced that CO concentration is a good tool to evaluate the quality of a measurement (similarly to "melt layer", see before), or the quality of a sample. CO can**

**be produced by chemical processes (the authors mention biological production of CO, citation is missing for that), but to my knowledge no collocated productions of CO and, e.g. $CO_2$, $CH_4$ or $N_2O$ have been reported so far in ice cores. The processes involved could be different, and a sample compromised for CO can be of good quality for others analyses. Ambient CO is often higher than what is in ice core bubbles, but this is also clear with $CH_4$.**

Author's response: High CO is quite a reliable indicator of contamination by the ingress of lab/storage facility air into the sample, or by leaks during processing the sample. High CO has also been reported together with high $CO_2$ and low $\delta^{13}C$-$CO_2$ values in a study of Antarctica vs Greenland $CO_2$ by Francey et al. (1997).

Author's changes in the Supplement: At page 2, we have cited Francey et al. (1997) showing evidence of in-situ production of CO in Greenland ice. We have modified the associated sentence: "*High CO values have been measured in Greenland ice (Francey et al., 1997) together with high $CO_2$ and low $\delta^{13}C$-$CO_2$, suggesting in situ oxidation of organic material deposited on the Greenland icecap (Francey et al., 1997)*".

---

## Author Comment (AC4) · 8 Mar 2019

Find the marked-up manuscript with track changes in the attached document

Please also note the supplement to this comment:
https://www.earth-syst-sci-data-discuss.net/essd-2018-146/essd-2018-146-AC4-supplement.pdf
* * *

---

## Author Comment (AC5) · 8 Mar 2019

[revised manuscript text omitted]

c. apply blank, gravity and diffusion corrections to measurements,

d. store and access information available from different types of measurements that are not part of the usual GASLAB/ICELAB analytical procedures (e.g $\delta^{18}O$, $H_2O_2$, etc...).

Finally, "Ice/Tests/Firn Output" are web-based interfaces that allow data extract based on multiple criteria.

**S1.1 Sample selection**

A rule-based method has been developed to allow for consistent and automatic sample flagging. The method separately considers the two parts where biases can be introduced:

1. Sample storage/preparation/extraction: flags have been associated with any possible source of uncertainty or bias. The diagnostic fields that have an associated flag are:

a. Fields describing visual characteristics, such as "crusts", "melt layers", "cracks", etc…

b. Fields recording the main factors influencing the extraction procedure, e.g.: "Pressure in the extraction line before water trap cooling", "Pressure in the extraction line after water trap cooling", "Temperature of cryogenic trapping", etc…

c. The measured CO concentration: carbon monoxide is used as a diagnostic tool. High CO values have been measured in Greenland ice (Francey et al., 1997) together with high $CO_2$ and low $\delta^{13}C$-$CO_2$, suggesting in situ oxidation of organic material deposited on the Greenland icecap (Francey et al., 1997)as it can be produced in-situ by chemical/biological reactions and is an indication of the quality of the sample. High CO concentrations can also indicate

contamination during sample storage/preparation/extraction as the ambient CO concentration is often higher than what it is in ice core bubbles.

  d. Fields having comments typed in by the operator. For these, the operator types in the flag value as well.

The flags are assigned integer values between 0 and 3, with 0 being no problem and 3 being evidence of a fatal problem. All flags are averages to provide a "quality score" (q.s.) and this number is used to set a "quality flag" (q.f.), according to the following thresholds:

- q.f.="reject" if q.s. = 3
- q.f.="poor" if q.s. > 0.5
- q.f.="fair" if $0.3 \leq$ q.s. $\leq 0.5$
- q.f.="good" if q.s. < 0.3

For example, a melt layer is classified as evidence of a fatal problem of sample quality and provides a q.s. = 3 and a q.f. = "reject".

2. Sample analysis: a weighting system has been used to attribute weights (w) to different replicated measurements of the same sample and to calculate a weighted average. (only forOnly a few measurements of concentration have been replicated., not f For isotopes, for which there is normallyhas never been not enough air to measure replicates). The weights are associated with:

  a. The volume injected into the instruments

  b. Any issue arising from the integration of the GC peak (baseline, shape, etc...)

  c. The difference between the results derived from peak area vs peak height integration.

The weights can get a value between 0 and 1 (0, 0.3, 0.5, 1), with zero being evidence of a fatal problem. A summary weight takes the minimum value of all weights, allowing us to be as conservative as possible, albeit at the cost of losing samples. A "measurement score" (m.s.) averages the summary weights of all replicates, providing one number which is then converted into a "measurements quality" (m.q.):

  - m.q.="poor" if m.s.< 0.4

  - m.q.="fair" if $0.4 \leq$ m.s. < 0.7

  - m.q.="good" if m.s. $\geq 0.7$.

The weighted average is calculated from the measured concentration ($x_i$) of each replicate and the corresponding summary weight ($w_i$) as: $x_w = \Sigma_i(x_i * w_i)/\Sigma_i w_i$ and provides the best estimate of the concentration measured for each sample. The difference between the average and the weighted average shows the impact of the weights on the averaging. In case the summary weight of a sample equals zero, only the arithmetic average is calculated (m.q.="reject").

While the flagging system refers to the quality of a sample, the weighting system refers to the quality of a measurement. Given that multiple (partially or totally) independent measurements can be performed on a sample (for different species such as $CH_4$, $CO_2$, CO and $N_2O$), there is a summary weight associated with each type of measurement carried out (m.s._{CH4}, m.s._{CO2}, etc...).

The flagging and weighting thresholds are tuned by calibrating the rule-based selection on the manual selection used before the database was conceived. The idea was to replicate with an automatic procedure what had been done in the past manually. The aim was to make the procedure consistent over time and as independent from the operator as possible. In summary, the rule-based selection converts qualitative judgments on the robustness of sample preparation, extraction and analysis into quantitative scores in order to consistently select/reject the results and quantify uncertainty.

**S1.2 Results corrections**

A number of corrections are applied to the measured concentrations and isotopic composition of trace gases. They are, following the order in which they are applied:

- The blank correction: this is related to any effect arising during sample storage/preparation/extraction. To quantify these effects, tests are run with (BFI) or without ice (Blanks). There is no reference ice core material, with known concentration and isotopic composition of trace gases available. We have chosen to simulate the extraction procedure is by growing gas-free ice (that is ice with no visible bubbles and is thus assumed to be gas-free) and inject a reference gas in the extraction line before starting the extraction procedure. This is in part also simulating the storage and the preparation because the BFI is trimmed at the same stage of processing and with the same band saw used for ice core samples, and is stored between preparation and extraction in the same conditions as ice core samples. The blank correction is quantified by the average deviation of replicated BFI/Blanks measured concentration and isotopic composition from the expected value (i.e.: the value associated with the reference gas used). A blank correction is calculated for each period when the conditions of preparation/storage/extraction are the same. In other words, a new blank correction is calculated each time any of the factors (operator, freezer, temperature of cold room, duration of extraction, etc...) that are believed to influence preparation/storage/extraction changes. The blank correction has an uncertainty associated with it, given by the standard deviations of differences from the expected value. The blank correction can vary significantly depending on the conditions of the extraction line and on how experienced the line operator is. Typical values are within the following ranges: 0.5-1.5 ppm (uncertainty 0.5-2 ppm, $1\sigma$) for $CO_2$, 3-10 ppb (uncertainty 3-15 ppb, $1\sigma$) for $CH_4$, 0.5-3 ppb (uncertainty 0.5-4 ppb, $1\sigma$) for $N_2O$ and 0.03-0.1 ‰ (uncertainty 0.04-0.13 ‰, $1\sigma$) for $\delta^{13}C$-$CO_2$.

- The gravity correction: Gravitational enrichment of heavier species in air in the firn open porosity (Craig et al., 1988; Schwander et al., 1988) has different effects depending on the difference between the mass of the measured species and the average mass of air. It can be evaluated as: $[X]_{corr} = 10^{-3} \times \delta^{15}N \times (M_X - M_{air}) \times [X]_{meas}$, where X is the measured species (i.e.: $CH_4$, $CO_2$ and $N_2O$), "corr" and "meas" stands for corrected and measured respectively, and $\delta^{15}N$ is the isotopic ratio of molecular nitrogen ($N_2$) in firn. For $\delta^{13}C$ measurements, the gravity corrected $\delta^{13}C$ equals the sum of a correction factor (very close to the $\delta^{15}N$-$N_2$) and the measured $\delta^{13}C$ (see Rubino et al., 2013 for details). The values of $\delta^{15}N$ are often measured in firn to constrain the firn diffusion model. In case of missing $\delta^{15}N$ measurements, the firn model is constrained with other measurements and then used to simulate the $\delta^{15}N$ profile. Our database stores all measured and modelled $\delta^{15}N$ values for firn sites at Law Dome. At Law Dome, the gravity correction is typically 1-1.5 ppm for $CO_2$, 2-6 ppb for $CH_4$, 1-1.4 ppb for $N_2O$ and 0.25-0.3 ‰ for $\delta^{13}C$-$CO_2$.

[revised manuscript text omitted]

Francey, R. J., Michel, E., Etheridge, D. M., Allison, C. E., Leuenberger, M. and Raynaud, D.: The pre-industrial difference in $CO_2$ from Antarctica and Greenland ice, in Fifth International Carbon Dioxide Conference, edited by R. Baum, I. Enting, R. Francey, M. Hopkins, and P. Holper, pp. 211–212, CSIRO, Division of Atmospheric Research, Melbourne, Cairns, Australia., 1997.

Joint Committee for Guides in Metrology: Evaluation of measurement data — Guide to the expression of uncertainty in measurement, Int. Organ. Stand. Geneva ISBN, 50(September), 134, doi:10.1373/clinchem.2003.030528, 2008.

Schwander, J., Stauffer, B. and Sigg, A.: Air mixing in firn and the age of the air at pore close-off, Ann. Glaciol., 10, 141–145, 1988.

Trudinger, C., Enting, I. G., Etheridge, D. M., Francey, R. J., Levchenko, V. A. and Steele, L. P.: Modeling air movement and bubble trapping in firn, J. Gephysical Res. - Atmos., 102(D6), 6747–6763, 1997.

Trudinger, C. M.: The carbon cycle over the last 1000 years inferred from inversion of ice core data, Monash University. [online] Available from: http://www.cmar.csiro.au/e-print/open/trudinger_2001a0.htm, 2000.

Trudinger, C. M., Enting, I. G., Rayner, P. J., Etheridge, D. M., Buizert, C., Rubino, M., Krummel, P. B. and Blunier, T.: How well do different tracers constrain the firn diffusivity profile?, Atmos. Chem. Phys., 13(3), 1485–1510, doi:10.5194/acp-13-1485-2013, 2013.

[Figure]

Figure S1: Screenshot of ICEBASE showing the general structure and the tables of the Icelab database.